# RNA cis-regulators are important for *Streptococcus pneumoniae in vivo* success

Indu Warrier[1,2], Ariana Perry[1¤], Sara M. Hubbell[1], Matthew Eichelman[1], Tim van Opijnen[3,4], Michelle M. Meyer[1]*

1 Boston College Department of Biology, Chestnut Hill, Massachusetts, United States of America, 2 RNA Therapeutics Institute, University of Massachusetts Chan Medical School, Worcester, Massachusetts, United States of America, 3 Broad Institute of MIT and Harvard, Cambridge, Massachusetts, United States of America, 4 Boston Children's Hospital, Division of Infectious Diseases, Harvard Medical School, Boston, Massachusetts, United States of America

¤ Current address: University of Massachusetts Chan Medical School, Worcester, Massachusetts, United States of America
* m.meyer@bc.edu

**Data Availability Statement:** All relevant data are in the manuscript and its supporting information files.

## Abstract

Bacteria have evolved complex transcriptional regulatory networks, as well as many diverse regulatory strategies at the RNA level, to enable more efficient use of metabolic resources and a rapid response to changing conditions. However, most RNA-based regulatory mechanisms are not well conserved across different bacterial species despite controlling genes important for virulence or essential biosynthetic processes. Here, we characterize the activity of, and assess the fitness benefit conferred by, twelve cis-acting regulatory RNAs (including several riboswitches and a T-box), in the opportunistic pathogen *Streptococcus pneumoniae* TIGR4. By evaluating native locus mutants of each regulator that result in constitutively active or repressed expression, we establish that growth defects in planktonic culture are associated with constitutive repression of gene expression, while constitutive activation of gene expression is rarely deleterious. In contrast, in mouse nasal carriage and pneumonia models, strains with either constitutively active and repressed gene expression are significantly less fit than matched control strains. Furthermore, two RNA-regulated pathways, FMN synthesis/transport and pyrimidine synthesis/transport display exceptional sensitivity to mis-regulation or constitutive gene repression in both planktonic culture and *in vivo* environments. Thus, despite lack of obvious phenotypes associated with constitutive gene expression *in vitro*, the fitness benefit conferred on bacteria via fine-tuned metabolic regulation through cis-acting regulatory RNAs is substantial *in vivo*, and therefore easily sufficient to drive the evolution and maintenance of diverse RNA regulatory mechanisms.

## Author summary

Regulation of gene expression is a nearly universal trait across all organisms that enables efficient utilization of resources, rapid reaction to a changing environment, or development of specialized subpopulations. However, the mechanisms that enable gene control are extraordinarily diverse. In bacteria, there are numerous regulatory mechanisms that

**Funding:** This work is supported by grants R01GM115931 to MM and TvO, and R01GM134259 to MM, from the National Institutes of Health, National Institute of General Medicine Sciences (https://www.nigms.nih.gov/). IW, AP, SH, TvO, and MM received salary from these grants. The funder had no role in the study design, data collection and analysis, decision to publish or preparation of the manuscript.

**Competing interests:** The authors have declared that no competing interests exist.

act at the RNA level. Such mechanisms are frequently responsible for fine-tuning expression levels of critical metabolic processes or virulence factors, but are often not well conserved across divergent bacterial species. In this work, we seek to evaluate the selective pressures driving the evolution and maintenance of RNA cis-regulatory mechanisms. Using the opportunistic pathogen *Streptococcus pneumoniae* as a model, we measured the fitness of strains carrying native locus mutations to RNA regulators that result in constitutively active and repressed gene expression. We find that only constitutively repressed mutants show significant growth defects under stringent *in vitro* culture conditions, but both constitutively active and repressed mutants are significantly less fit within mouse models of infection and carriage. We also find that specific pathways such as pyrimidine and FMN synthesis/transport are very sensitive to mis-regulation and constitutive repression respectively. This study demonstrates that while individual variants may show mild fitness defects, when considered in aggregate, these defects correspond to significantly depressed fitness values that are more than sufficient to drive evolutionary trajectory for organisms with large population sizes such as *S. pneumoniae*. Thus, RNA-regulators responsible for fine-tuning metabolic genes are likely critical for *S. pneumoniae* success in a host.

## Introduction

Modern sequencing approaches have illuminated the transcriptional landscape of many bacterial pathogens [1,2] revealing the extent to which diverse types of RNA regulatory elements are involved in controlling gene expression after initiation of transcription. RNA regulators control genes involved in primary metabolism, virulence [3–5] and stress responses [6] suggesting that they play key roles in bacterial physiology and adaptation to diverse environments including antibiotic treatment [7,8]. Cis-acting RNA regulators are canonically located within the 5'-UTR of the regulated transcript and include a structured portion that acts as a sensor to detect temperature [9] or various cellular components including small molecules [10], tRNAs [11], or proteins [12]. An array of different mechanisms translate ligand binding into changes in gene expression. These include transcription attenuation [13], control of translation initiation [14], and regulation of transcript stability [15]. Due to their association with essential biosynthetic pathways, such regulators are touted as potential antimicrobial targets [16].

Despite the association of RNA regulators with key bacterial processes, they are not always well-conserved across large phylogenetic distances, and it is not necessarily clear how important such regulatory mechanisms are for organismal fitness in natural environments or how strong the selective pressure to maintain regulation may be. Relatively few bacterial noncoding RNA (ncRNA) families are broadly distributed [17] with cis-acting RNAs typically identified across broader ranges of species than trans-acting RNAs [18]. Most RNA regulators are found in only a few genera [19], suggesting either relatively recent origins [20], frequent loss [21], or a combination of both. The structural complexity of an RNA cis-regulator is not necessarily correlated with its distribution, as even complex structures are often found in only narrow bacterial lineages [22]. Furthermore, well-conserved aptamer ligand-recognition domains may recognize the same effectors to trigger diverse mechanisms of gene regulation in different organisms.

Phenotypes associated with the loss of any one RNA regulator are often not observed, or are very subtle [23,24]. This lack may be due to the relatively small dynamic range of many

such regulators [25], redundancy of function for trans-acting regulators [26,27], or lack of assessment in the appropriate environment. Models have suggested that benefits of trans-acting sRNA networks include more graduated responses [28], rapid adaptation to large input signal changes [29,30], and noise reduction [31,32], and the role of negative feedback in biological systems seems to be to accelerate a return to equilibrium [33]. All of these features may be challenging to measure in planktonic culture, which does not typically mimic the intrinsically dynamic environment in which most microorganisms have evolved to survive. More generally, the benefit of regulatory activity in comparison to a constitutively expressed protein is often rationalized in the context of metabolic cost [34], and is predicted to be relatively small for most cellular proteins [35,36]. When regulatory RNAs are directly targeted by antimicrobials, resistance mutations arise that break the regulatory capacity of the RNA resulting in constitutive gene expression [37–40], suggesting that the selection pressure to maintain regulation may not be strong.

The contributions of individual genes to organismal fitness have been assessed on a genome-wide basis via pooled competition assays such as transposon insertion sequencing [41], and the fitness cost of constitutive gene expression has largely been assessed in the context of over-expression libraries [42–46]. However, specifically targeting regulatory sequences to assess the impact of both over- and under-expression remains challenging to accomplish in a high-throughput manner. In past work we have found that mutations to individual RNA cis-regulators displaying either constitutively active and repressed gene expression confer phenotypes such as cold sensitivity or defects in motility and biofilm formation in *Bacillus subtilis* [47,48]. However, these experiments did not enable extrapolation to a broader context due to the small number of examples examined, and the limited ability of culture conditions to reflect natural environments of *B. subtilis*.

In this work, we strive to more broadly assess the selective pressures on cis-acting RNA elements by examining a series of cis-regulatory RNAs using *Streptococcus pneumoniae* TIGR4 as a model. *S. pneumoniae* is an opportunistic pathogen carried asymptomatically by 20–90% of children under 5 [49] whose native environment is the human nasal cavity [50], and which may cause pneumonia, otitis media, or invasive pneumococcal disease (meningitis or bacteremia) [51]. Due to rising antibiotic resistance [52–54], and serotype switching in response to available vaccines [55–57], *S. pneumoniae* remains a pathogen with important public health implications [58]. Clinical *S. pneumoniae* isolates exhibit a large pangenome consisting of a core genome of ~2100 genes common to all strains, and an accessory genome of over 4000 genes found in subsets of strains [59]. Thus genomic analysis to leverage the pangenome [59–62] and animal infection models [63], play critical roles in understanding *S. pneumoniae* physiology. Therefore, *S. pneumoniae* represents a good model in which to assess selective pressure via both comparative genomic and experimental methodologies.

Here, we characterize a set of twelve well-conserved RNA cis-regulators controlling genes associated with the biosynthesis or transport of essential nutrients in *S. pneumoniae*. We go on to create strains with targeted mutations resulting in constitutive activate or repressed gene expression. Our assessment of these strains during planktonic growth in culture, and within mouse infection models that mimic nasal colonization, invasive pneumococcal lung infection, and transition to blood upon lung infection, shows that both constitutive activation and repression of gene expression are deleterious to bacterial fitness within *in vivo* environments. Our findings suggest that the impact of RNA-based gene-regulation in a more realistic environment is substantial and even such fine-tuners of gene expression play critical roles in mediating success in dynamic environments.

## Results

### Assessed RNA regulators are highly conserved across *S. pneumoniae* strains and repress gene expression in response to ligand binding

To assess the role of RNA cis-regulation in *S. pneumoniae* physiology, we examined six structurally distinct RNA cis-regulator classes. In several cases we investigated multiple instances of the same regulator class throughout the genome, with each instance controlling different gene sets (S1 Fig). The set of regulators included two pyrR RNAs that are responsive to a uracil monophosphate (UMP)-bound PyrR protein, a guanine riboswitch responding directly to guanine, two FMN riboswitches responsive to the cofactor flavin mononucleotide (FMN), four TPP riboswitches responsive to the cofactor thiamine pyrophosphate (TPP), a glycine riboswitch, and a T-box RNA responsive to tryptophan-tRNA. When combined with the additional pyrR RNA we assessed in a previous work [64], this yields a total of twelve individual regulatory regions. These regulators were chosen for their degree of conformity with existing RNA primary and secondary structure consensus models [22,65], and the availability of extensive biochemical characterization or high-quality structural data for one or more homologs to support selection of binding site mutations [66–72].

To assess the extent to which the chosen regulators are conserved across the *S. pneumoniae* pangenome, we identified the sequences corresponding to each regulator and its regulated genes across a panel of genomes for a representative set of *S. pneumoniae* strains [73] (S1 Table). All of the RNA regulators chosen for study, and their associated regulated genes, are part of the *S. pneumoniae* core genome [62,73]. All of them regulate operons including genes coding for the transport (n = 5), biosynthesis (n = 5), or a combination thereof (n = 2), of critical metabolites (S1 Fig). They are highly conserved across the set of representative *S. pneumoniae* clinical strains, displaying little sequence variation (see S1 Table for sequence accession numbers). Several regulators display no nucleotide variation (glycine and guanine riboswitches), while others show variation at only a few positions. The average identity to the regulator sequence in *S. pneumoniae* TIGR4 is 98.8% over all 11 regulator sequences assessed, which corresponds to approximately 0–4 mutations per regulator across all comparisons. This level of identity is similar to that of the downstream protein-coding genes (average 99.4% nucleotide identity to the *S. pneumoniae* TIGR4 sequence). Thus, the set of regulators chosen for analysis here are within the *S. pneumoniae* core genome, and highly conserved.

While each regulator is a homolog of a well-established RNA cis-regulator, none have been previously examined in *S. pneumoniae*. To establish the behavior of these regulators in *S. pneumoniae*, we measured gene expression in the presence and absence of the ligand or ligand precursor. While most of the regulators are predicted to utilize premature transcription termination mechanisms (S2 Fig), these are putative unverified structures. Thus, we assessed gene expression using translational reporters that fuse a β-galactosidase reporter gene with the native regulatory region (including the native promoter identified based on experimentally determined transcription start sites [64] and the first 3–4 codons of the proximal regulated gene). As we added the β-galactosidase reporter to a common neutral locus of *S. pneumoniae* TIGR4 for all reporters, all biosynthetic pathways are intact in the reporter strains.

The gene expression of each reporter was confirmed by assessing β-galactosidase activity in a chemically defined medium (CDM) (S2 Table) in the presence and absence of respective ligand or ligand precursor (see methods). All of our regulators are genetic OFF switches that reduce gene expression in the presence of their respective ligands (Fig 1A). Most RNAs displayed between 3.5- and 12-fold repression, but the extent of regulation varies from over 100-fold, to approximately 2.5-fold. Our results are consistent with those observed in other organisms where response profiles of individual RNA regulatory mechanisms may be quite

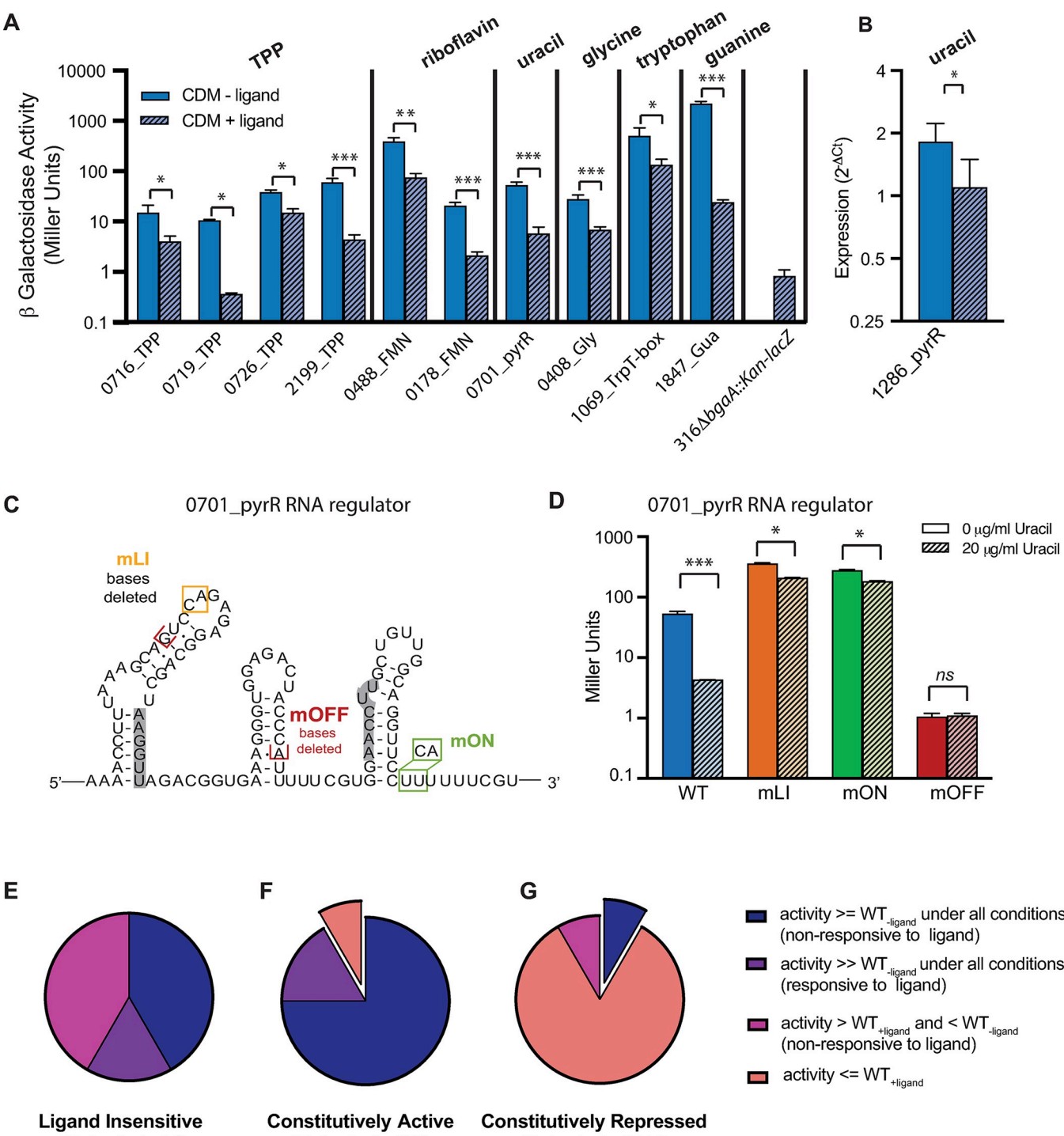

**Fig 1. *S. pneumoniae* RNA regulators are ligand responsive and most designed mutations function as anticipated. (A)** β-galactosidase activity of RNA reporter strains grown in the absence and presence of ligand or ligand precursor (0.3 μg/mL thiamine, 140 ng/mL riboflavin, 20 μg/mL uracil, 75 μg/mL glycine, 204 μg/mL tryptophan, 50 μg/mL guanine) demonstrates regulation of gene expression with a significant difference between expression in the two conditions. Bar height indicates the mean replicate value, error bars indicate standard error of the mean, individual points shown in S2 Fig. Significance assessed via an unpaired two-tailed t-test, * p<0.05, ** p < .001, ***p < .0001. **(B)** qRT-PCR of SP_1286 transcript was used to quantify expression in response to ligand (due to exceeding low expression from the native promoter for SP_1286). Bar heights represent mean value of biological replicates, error bars and statistical analysis as above. **(C)** Illustration of regulatory mutants using 0701_pyrR regulator as an example. mLI, mON, and mOFF mutations are indicated on a putative secondary structure. **(D)** Behavior of 0701_pyrR regulatory mutants demonstrated via β-galactosidase assay (Miller Assay) [94]. Both mLI and mON mutants show increased gene expression compared to WT, but remain somewhat ligand responsive (significant decrease in the presence of ligand, *p<0.05).

**(E)** Behavior of RNA regulator mutants designed to be ligand insensitive (mLI) suggest that they have constitutive gene expression (S2 Fig). Their behavior may be categorized as: 1) ligand non-responsive and greater than or equal to the unrepressed WT activity (WT$_{-ligand}$) under both conditions; 2) ligand responsive and much greater than the unrepressed WT activity under both conditions; or 3) ligand non-responsive with activity between the repressed (WT$_{+ligand}$) and unrepressed (WT$_{-ligand}$) WT activities. **(F)** Behavior of RNA regulator mutants designed to have constitutive gene expression (mON). Their behavior may be categorized as above, with one exception which shows lower expression than the repressed WT (WT$_{+ligand}$) (S2 Fig). This mutant (exploded pie slice) was removed from further studies. **(G)** Behavior of RNA regulators designed to have constitutively repressed gene expression. Categorization as above. One mutant is slightly elevated compared to the unrepressed WT and is thus between the repressed and unrepressed activities, and we retained this mutant. One mutant shows higher expression than the unrepressed WT (exploded pie slice) and was removed from further studies (S2K Fig). All numeric data points in S1 Data.

variable [74]. Furthermore, our results likely reflect the biological function of such RNAs as tuning mechanisms more than absolute ON or OFF switches [75]

For one of the regulators assessed, pyrR RNA preceding SP_1286 (1286_pyrR), no β-galactosidase activity was detectable from the reporter construct suggesting extremely low expression under the conditions examined. As this pyrR RNA incorporates a putative intrinsic terminator, we utilized qRT-PCR to directly assess transcript abundance in order to evaluate gene expression for this regulator (Fig 1B). We found that gene expression was repressed by approximately 2-fold in cells grown with uracil. Comparison with the control transcript in qRT-PCR, and existing *S. pneumoniae* TIGR4 RNA-seq data collected under similar conditions [64], confirm the very low expression of the native transcript under both the growth conditions examined here, suggesting that our low β-galactosidase activity is attributable to a weak native promoter under these culture conditions.

## Designed mutations to RNA regulators yield predicted changes in gene expression

We next designed three different mutations (mLI, mON, and mOFF) for each RNA regulator that impact the predicted secondary structure of the RNA in order to disrupt its regulatory activity (an example RNA is shown in Fig 1C, others illustrated in S2 Fig). The mLI–ligand insensitive mutations are designed based on available functional and structural data [66,68–72] to disrupt binding of the respective ligand to the RNA via one or two nucleotide changes or deletions. The mON mutations are designed to disrupt the RNA folding resulting in constitutive gene expression, "ON". The change may be a deletion or mutation to destabilize or disrupt a premature intrinsic terminator stem, or truncate the aptamer leaving only a viable ribosome binding site. Individual mON mutations varied significantly depending on the putative mechanism of the individual RNA regulator. The mOFF mutations are designed to result in constitutively repressed gene expression, "OFF", and typically delete the aptamer region leaving only an intrinsic termination stem, or mutate a predicted Shine-Dalgarno sequence to prevent ribosome binding.

To confirm the modified behavior for each mutation we used the β-galactosidase reporter system described above, incorporating each mutation. In the case of 1286_pyrR, we used qRT-PCR of strains with native locus mutations. Consistent with our findings that all of the regulators examined are OFF switches, the ligand insensitive mLI mutations typically display constitutive β-galactosidase activity that is equal to or exceeds the activity of the unrepressed (-ligand) WT constructs (n = 7)(S2 Fig). In some cases, the activity is substantially elevated compared to the unrepressed WT activity (5 to 10-fold increase) in both the presence and absence of ligand, but remains somewhat ligand responsive (n = 2). An example of a Miller Assay to measure β-galactosidase activity is highlighted in Fig 1D, where the mLI mutant of 0701_pyrR remains ligand responsive, but is expressed at a much higher level than the WT regulator. Finally, many of the mutants show activity that is unresponsive to ligand, but falls between the repressed and unrepressed WT activity (n = 5) in both conditions (Figs 1E and S2).

The majority of mON mutants display constitutively active gene expression with β-galactosidase activity greater than or equal to the unrepressed WT (n = 9), and most of the remaining strains display high β-galactosidase activity that is still somewhat ligand responsive as described above (n = 2) (Figs 1F and S2). The mOFF mutants are typically constitutively repressed with activity less than the repressed WT activity, regardless of ligand presence or absence (n = 10), with one mutant showing activity slightly greater than the repressed WT, but less than unrepressed WT (Fig 1G, [64]).

For both the mLI and mON mutants the increased gene expression observed is likely due to stabilization of the transcript in a form where read-through of the intrinsic terminator is favored. We, and others, have observed in 3'-sequencing studies that many riboswitches alter the extent of read-through for intrinsic terminator stems, and it is rarely 0% or 100% utilization for such conditional terminators [7,64]. Thus, it is entirely possible that our mutation may stabilize an anti-terminator stem to a greater extent than it would be in the absence of ligand of a wild-type strain. Furthermore, many mON mutants completely remove the intrinsic terminator. We speculate that the retained ligand responsiveness may derive from promoter level regulation of transcription initiation, which is unperturbed in our mutants.

There are two examples where the mON or mOFF mutations did not display the anticipated activity (Fig 1F and 1G). The 0726_TPP_mON mutant approximates a repressed phenotype rather than a constitutively ON phenotype (S2C Fig). For 1286_pyrR, we find the 1286_pyrR_mOFF mutant displays expression levels substantially higher than the unmutated sequence (S2K Fig). Based on these assessments of gene expression, we carried forward with all our mutants except these two anomalies. In the case of 0726_TPP_mON, the ON phenotype is displayed by 0726_TPP_mLI. In the case of 1286_pyrR, the measured expression for the native transcript is so low that evaluating a strain with a further reduction to establish reduced capacity would be very challenging. Furthermore, transposon sequencing studies in *S. pneumoniae* TIGR4 suggest that SP_1286 is dispensable within *in vivo* environments [76].

## Constitutive repression of gene expression is deleterious for growth in medium lacking target nutrients

To assess the effect of RNA regulation on *S. pneumoniae* fitness we created a set of strains for each regulator where the native locus is replaced with each mutant (mLI, mON, and mOFF). To enable selection of the mutants following homologous recombination a Chl[r] cassette is inserted upstream of the native promoter (Fig 2A). We also generated a matched control strain for each regulator containing only the Chl[r] cassette and no changes to the regulatory sequence (WT[c]). To determine the effect of RNA regulator mutations on *in vitro* growth, we assessed the growth of each mutant in CDM in the presence and absence of the appropriate ligand or ligand precursor.

The individual mutant strains displayed a range of phenotypic defects from severe to undetectable (S3 Fig). To quantify the *in vitro* growth characteristics of each strain, the doubling time during exponential phase and carrying capacity (as measured by maximum $OD_{600}$) were extracted from the curves [77]. Both of these parameters ultimately contribute to bacterial fitness; a strain with a longer doubling time (or lower growth rate where growth rate is = ln(2)/doubling time) in the absence of population size restrictions will be less fit, and a strain with a lower carrying capacity will ultimately have fewer numbers in an environment where resources are limited. We find that most strains showed no significant changes in either carrying capacity or growth rate compared to the WT[c] strains in either of the culture conditions assessed (Fig 2B and 2C). The majority of the significant changes are decreases in growth rate or carrying capacity and strains that display decreases are most likely to be the mOFF strains with

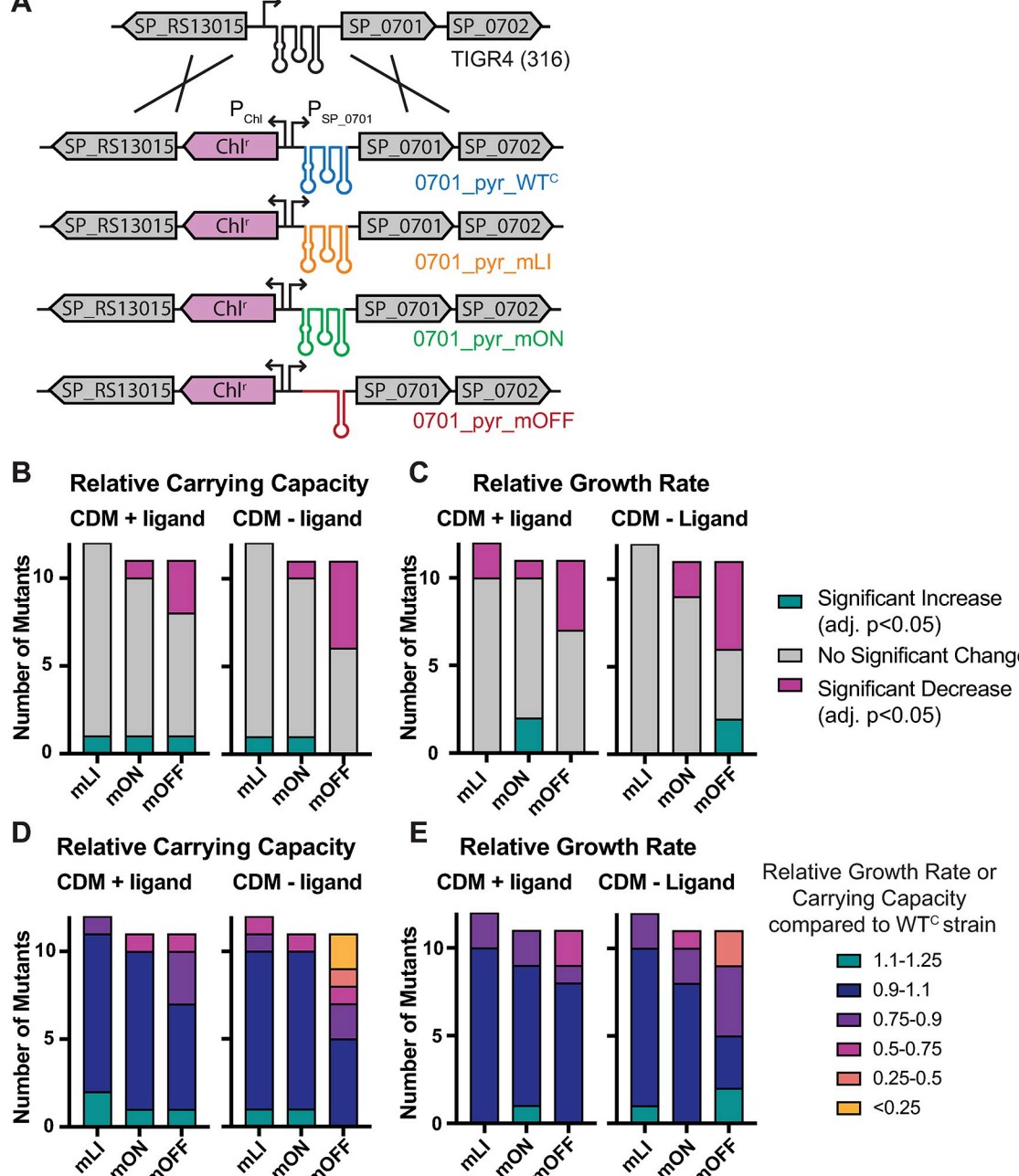

**Fig 2. Mutations to *S. pneumoniae* RNA regulators that repress gene expression result in inhibited growth in culture. (A)** Schematic illustrating the strains constructed to assess impact of the 0701_pyrR regulator on organismal fitness as an example of the process. Mutations were integrated into the native locus accompanied by a chloramphenicol (Chl$^r$) marker. A WT$^C$ strain including just the Chl$^r$ marker at the same position was also created as a control. This approach was taken for each RNA regulator in this study at each respective locus. **(B)** Enumeration of each type of mutant (mLI, mON, mOFF) with statistically significant (adj. p<0.05) changes compared to the WT$^C$ control in relative carrying capacity in CDM +/- target ligand (S3 Fig). **(C)** Enumeration of each type of mutant with statistically significant changes in relative growth rate in CDM +/- target ligand (S3 Fig). **(D)** Relative carrying capacity of each type of mutant in CDM +/- target ligand (S3 Table). **(E)** Relative growth rate of each type of mutant in CDM +/- target ligand (S3 Table). All numeric data points in S1 Data.

constitutively repressed gene expression. Decreases in carrying capacity and growth rate are more prevalent in CDM lacking the target nutrient (-ligand condition) than in CDM +ligand (Fig 2B and 2C).

To better compare the magnitude of changes observed across the medium conditions assessed, we normalized the measurements for each mutant under each condition to the corresponding WT$^C$ strain to obtain relative growth rates and relative carrying capacities (Figs 2D, 2E, and S4D–S4G and S3 Table). From this analysis we observe that defects are more severe in CDM lacking ligand (- ligand), and that defects in relative carrying capacity are larger in magnitude than relative growth rate defects (the lowest relative growth rate is 0.3, while the lowest relative carrying capacity is 0.015, compared to WT, where WT = 1). While a few significant increases in growth rate or carrying capacity are observed, they are smaller in magnitude than the decreases (Fig 2D and 2E).

From assessing planktonic growth of our collection of mutants in a nutrient limited medium, we observe that most growth decreases occur in mOFF strains in the absence of the target ligand. Most of the strains displaying severe defects (relative growth rate or carrying capacity < 0.5) are constitutively repressed portions of the biosynthetic pathway for the targeted nutrient (S3 Fig). The associations between regulation of biosynthetic genes, lower carrying capacity, and rescue via nutrient supplementation suggests that the observed defects result directly from nutrient deficiencies.

## Both constitutive activation and repression of gene expression decrease fitness during infection

To determine the impact of RNA regulators on *S. pneumoniae* fitness in a more realistic and dynamic environment, we examined how mutations to nine of the regulators impact fitness during mouse nasopharynx colonization and lung infection. We omitted three of the four TPP riboswitches (0716_TPP, 0726_TPP, and 2199_TPP) due to the somewhat redundant gene sets regulated by the four different riboswitches. To determine the relative fitness of RNA regulator mutations during infection, we performed 1:1 competition assays between the parental strain (unmodified *S. pneumoniae* TIGR4) and each mutant in two mouse models: a nasopharynx colonization assay mimicking *S. pneumoniae* colonization and carriage where nasopharyngeal lavage is collected at 24–48 hours after infection, and a lung infection assay mimicking the *S. pneumoniae* invasive disease of pneumonia where the lung is collected 6–24 hours after infection. To assess vascular invasion following lung infection, blood was also collected from animals following the lung infection assay. From the initial inoculum and each collected sample, mutant (Chl$^r$ mutant strain) and total CFUs (mutant and parental strain) were enumerated to allow calculation of the relative frequencies of each strain at both the start and end of the experiment. Thus, the fitness (*W*) of the mutant strain relative to the reference strain can be determined (see methods) [78].

The outcome of the competition assay is a relative fitness value *W*, where *W* = 1 indicates that the mutant and the parental strain are equally fit with similar survival and growth rates within the environment assessed; *W*<1 indicates a mutant that is less fit than the parental strain. In an *in vivo* environment, a fitness defect may be caused by any number of issues including a slower relative growth rate, an inability to achieve part of the infection cycle (e.g. attachment or invasion), or more rapid clearance by the immune system. As a control comparison, alongside each set of mouse infections we performed a similar *in vitro* competition experiment in rich medium with a sample of the same inoculum. The fitness of the mutant strains determined from the *in vitro* rich media approximate 1 (average *W* = 1.00 +/- 0.077 across all experiments) (S5 Fig and S4 Table), indicating no inherent fitness defect in rich medium for

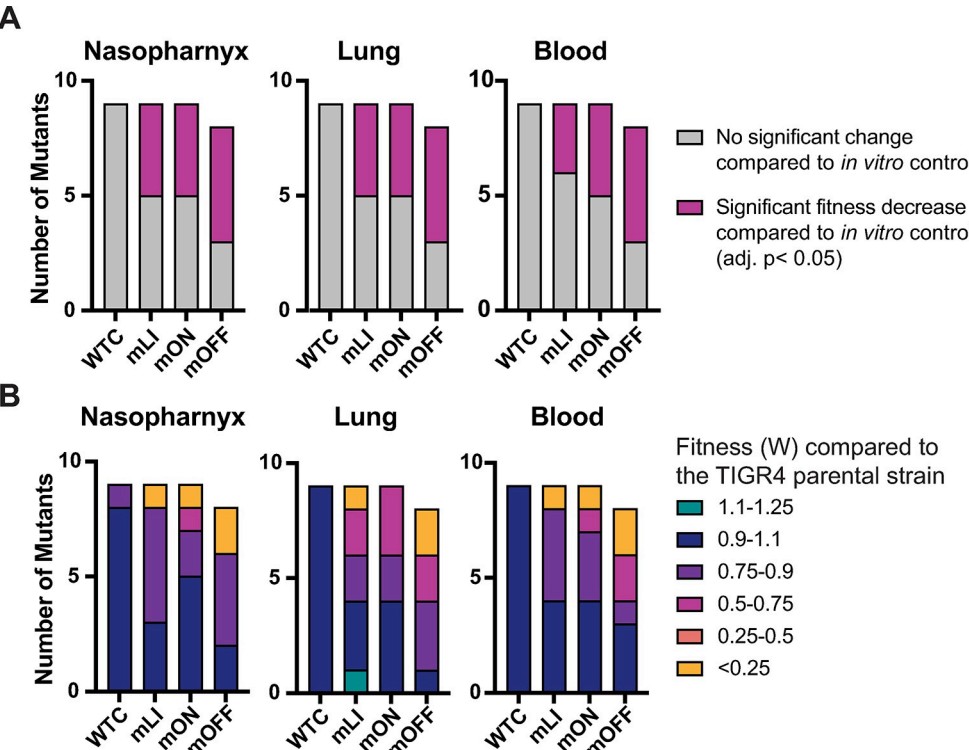

**Fig 3. Mutations to *S. pneumoniae* RNA regulators reduce fitness during mouse infection. (A)** Number of mutant strains that display a significant change in fitness for each type of mutant (mLI, mON, mOFF, and WT<sup>C</sup> control) in each *in vivo* environment (nasopharynx colonization, lung infection, transmission to blood) compared to *in vitro* control. Fitness determined via competition assays between strains carrying mutations to RNA regulators and the parental strain (S5 Fig). **(B)** Average fitness values observed for each class of regulator mutant in each environment (S4 Table and S4A–S4C Fig). All numeric data points in S1 Data.

our strains. We also assessed the WT<sup>C</sup> strain for each of our regulators to ensure that introduction of the Chl<sup>r</sup> marker alone did not introduce a fitness defect or advantage during infection, and we find that these fitness values are also close to 1 (average $W = 0.98$ +/- 0.061 across all experiments), and none display a significant fitness defect under any condition, indicating that our strains reflect fitness costs incurred by the change in regulatory activity rather than insertion of the antibiotic marker alone (S5 Fig and S4 Table).

In marked contrast to our findings in planktonic growth, many mutants with both constitutively active (mON) and repressed gene expression (mOFF) displayed significant decreases in fitness compared to the *in vitro* control (Figs 3A and S5). In any given *in vivo* environment, approximately half of all strains displayed a significant fitness decrease, and no significant fitness increases were observed. While one strain (0178_FMN_mLI) does show $W = 1.1$ in the lung (Fig 3B, lung and S4 Table), this finding is not statistically significant compared to the *in vitro* control ($W = 1.06$). In addition, a much greater proportion of the mutants showed larger decreases in relative fitness, with strains carrying each type of mutation (mLI, mON, and mOFF) displaying severe fitness decreases ($W < 0.25$)(Figs 3B and S4A–S4C). We found that for every RNA assessed, at least one mutant (mLI, mON, or mOFF) showed a statistically significant fitness decrease under at least one infection environment compared to the *in vitro* control (S4A–S4C and S5 Figs). Furthermore, 20 of 26 mutant strains displayed a statistically significant fitness decrease in at least one *in vivo* environment (S4A–S4C and S5 Figs, [64]).

Therefore, our infection studies suggest that *S. pneumoniae* fitness is negatively impacted by constitutive activation or repression of gene expression. Furthermore, approximately 75% of our mutant strains display statistically significant fitness defects in one or more *in vivo* environments, including several mutations with both constitutively active and repressed gene expression which induced severe fitness defects (W<0.25). This suggests that in some cases regulation is not merely beneficial but critical for success during infection.

## Aggregation of data across regulators demonstrates that mis-regulation is deleterious *in vivo*, while only repression of gene expression is deleterious in planktonic culture

To further assess the behavior of constitutively active or repressed gene expression across our collection of regulators as a whole, we aggregated individually measured values for all mutants of a particular type (mLI, mON, mOFF) across each growth condition and measurement (relative growth rate, relative carrying capacity, or fitness in each *in vivo* environment). For this analysis we equalized the number of measurements per strain in our assessed population via random sampling of three data points for each strain, repeating the sampling 100 times to ensure robustness (S5 Table).

By comparing the planktonic growth parameters of strains with each mutant type, we find that in aggregate, populations composed of the mOFF mutants are much more likely to display significant defects compared to WT$^C$ strains (Fig 4A and 4B). The growth rate in the -ligand condition is the most affected, with 83/100 samplings showing a significant difference (adj. p<0.05) from the WT$^C$ population, with an average adj. p-value of 0.034. For both the growth rate in the +ligand condition, and the carrying capacity in the–ligand condition a majority of the populations (62% and 69%) had p<0.05, but the mean adj. p-value in each of these cases was > 0.05 (S5 Table). Notably, only 4% of the mLI populations of growth rate in CDM+ligand (Fig 4B) showed a significance difference from WT$^C$, and the mLI and mON mutants showed no detectable phenotype in the remaining conditions. Thus, in aggregate we find that only constitutive repression of gene expression (mOFF) results in significant growth deficiencies, and these affects are more apparent in CDM lacking the ligand or ligand precursor.

Turning to the *in vivo* environments we observe a very different scenario. By comparing the *in vivo* fitness measurements of strains with each mutant type, we see that the populations of mLI and mOFF mutants are significantly less fit than the WT$^C$ population in all *in vivo* environments, and not the *in vitro* culture in rich medium. The mON mutants are significantly less fit than the WT$^C$ in the pneumonia and transition to the blood models (lung and blood, average adj. p-values = 0.0035 and 0.011 for sampled populations respectively) but not in the nasopharynx colonization model (average adj. p-value = 0.091, adj. p-value < 0.05 in only 48/100 of sampled datasets) (Fig 4C and S5 Table).

Thus, while we observed significant defects for only constitutively repressed gene expression mutants (mOFF) in culture, we find that in the context of infection, mutations that cause constitutive activation of gene expression are also generally deleterious. However, the fitness decrease is still stronger for constitutively repressed mutants than constitutively active mutants (average fitness of mOFF strains within *in vivo* environments is 0.56, while average fitness of mLI and mON strains is 0.70, and 0.73, respectively).

## Growth rate and carrying capacity measured in planktonic monoculture do not significantly predict *in vivo* behavior

We have demonstrated that strain fitness within *in vivo* environments is more sensitive to mis-regulation compared to growth in planktonic culture. However, we were curious to what

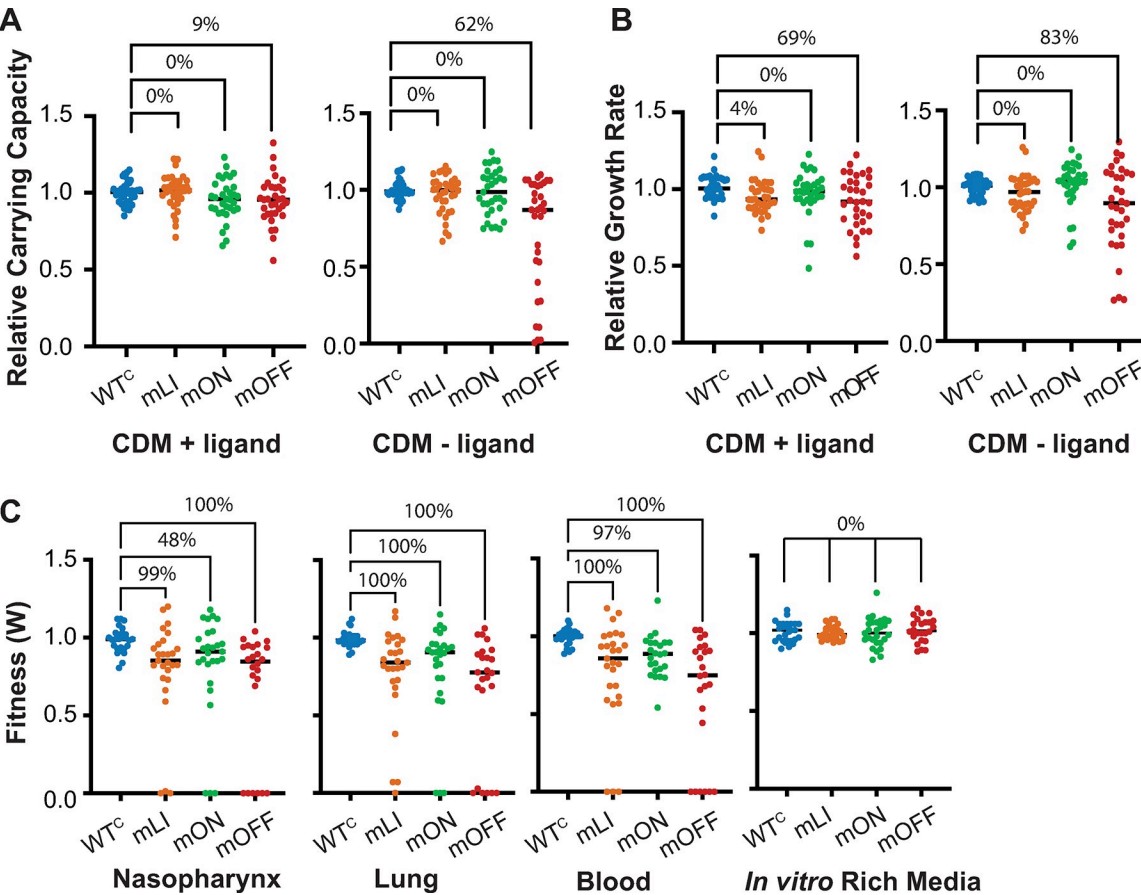

**Fig 4. Aggregation and sampling of individual data point across all regulators shows distinction between *in vivo* and *in vitro* environments.** (**A**) Aggregated and sampled relative carrying capacity across all regulators (S3 Fig). A randomly chosen sampled population is depicted. A total of 100 random samplings of the data were performed and the percentage of sampled populations displaying a statistically significant difference between matched control constructs (WT[C]) and mutant strains is indicated (Kruskal-Wallis followed by Dunn's test of multiple comparison, % of samplings where adj. p<0.05). Average adj. p-values and percentage of sampled populations where adj. p <0.05 reported in S5 Table. (**B**) Sampled and aggregated relative growth rates across all regulators (S4 Fig). (**C**) Sampled and aggregated mouse infection relative fitness values across the regulators examined (S5 Fig) in each *in vivo* environment (nasopharynx, lung, and blood) and the control conducted *in vitro* in rich medium. Average adj. p-values and proportion of samples where adj. p <0.05 are in S5 Table. All numeric data points in S1 Data.

extent the *in vivo* environments simply amplified existing growth rate differences between strains, or whether the different environments favor distinct sets of mutants. The fitness value (*W*) we determined from our *in vivo* competitions should be approximately equal to the relative growth rates under exponential phase growth. A qualitative comparison of relative growth rates, relative carrying capacity, and relative fitness across all our strains shows that these traits do not appear well correlated (S4 Fig). There is little overlap between strains showing the most severe defects *in vitro* compared to *in vivo* environments, and intuitively, our fitness values measured *in vivo* may also incorporate a range of other factors that planktonic culture cannot capture.

To more quantitatively compare *in vitro* growth with *in vivo* fitness measurements, we used a simple model to calculate the expected fitness of our strains given their relative growth rates and carrying capacities determined via *in vitro* planktonic monoculture [79]. Our model is a two-species logistic growth model that incorporates no interactions between the species, but does include the total population as a shared parameter that impacts growth rate as the

carrying capacity is approached (see methods). Using this model to calculate fitness in a competition ($W_{model}$) as a function of time over a range of parameter sets shows that there are typically two potential stable values (S6A Fig). As anticipated, the first value occurs while both strains are growing exponentially and corresponds to the relative growth rates of the two strains [80]. This equilibrium is disturbed as the carrying capacity of one strain is approached and its growth slows compared to the other. The second stable value is reached as the carrying capacity of both strains is reached and the population is no longer growing in stationary phase.

From our modelling of expected fitness based on monoculture parameters, we observe that the modelled fitness never approaches the very severe fitness defects ($W < 0.1$) observed for several strains during infection, even strains displaying severe defects in culture (S4A–S4C, S6B, and S6C Figs). By directly comparing our modelled and measured fitness values, we find the strongest correlation is between modelled exponential fitness (approximately equal to the relative growth rate) and measured fitness in the lung (Figs 5A and S7A–S7F). While significant, this correlation ($R^2 = 0.59$) is weak compared to the correlations between *in vivo* fitness values from different environments (S7G–S7I Fig, $R^2 = 0.84, 0.88,$ and $0.95$). The finding that exponential fitness more accurately predicts *in vivo* fitness is in agreement with our expectation given past estimates of growth parameters from infections (doubling time of 108 minutes in the nasopharynx or 161 minutes in the lung, high carrying capacity of the reference strain (as suggested by [76]), and 24–48 hours of growth). However, modelled exponential phase relative fitness is ultimately a poor predictor of measured fitness. In aggregate, our results indicate that modelled fitness during exponential phase (relative growth rate) is at best weakly correlated with measured *in vivo* fitness, while fitness measured under different *in vivo* conditions are significantly correlated (S7 Fig).

Given the lack of strong correlation between our modelled and measured fitness values, we experimentally tested the validity of our modelling by performing *in vitro* co-culture competition assays under conditions where the modelling predicts that we will observe fitness differences based on the planktonic growth parameters. Our dataset includes two strains that display growth rate and carrying capacity defects in CDM lacking uracil, 1278_pyrR_mOFF and 0701_pyrR_mOFF (S3G and S3L Fig [64]). We performed competitions between each of these strains, as well as their corresponding WT[C] strains, with the parental reference strain (*S. pneumoniae* TIGR4) in CDM +/- uracil. Based on the fitness modelling we expected both 1278_pyrR_mOFF and 0701_pyrR_mOFF to show fitness defects in the absence of uracil (Fig 5C and 5E). However, we observed that only 0701_pyrR_mOFF grown in the absence of uracil displays a significant fitness disadvantage compared to the parental strain in culture (Fig 5B and 5D). Control experiments using the WT[C] strains for each regulator showed no fitness disadvantage (S6D Fig)

In examining these findings, we note that 0701_pyrR_mOFF has a more severe defect in carrying capacity and a growth rate defect that is not completely rescued in the presence of uracil (+ligand), while 1278_pyrR_mOFF displays no phenotype in the presence of uracil (S3G and S3L Fig, [64]). Thus, we speculate that 1278_pyrR_mOFF growth is rescued in co-culture by cross-feeding from the reference strain, similar to the way its growth rate is rescued by the addition of uracil. In contrast, 0701_pyrR_mOFF is not completely rescued by the addition of uracil to the medium and thus is potentially less affected by the co-culture. We also observe that all fitness values measured during this experiment are slightly higher than anticipated based on the modelling. However, the general trends observed from our experiment match those expected from the modelling and potentially suggest that co-culture fitness as we are measuring it is a less sensitive metric than parameters extracted from monoculture growth curves due to potential cheating behavior wherein the mutant strain, which cannot synthesize pyrimidines itself, obtains sufficient amounts from the co-cultured wild-type strain.

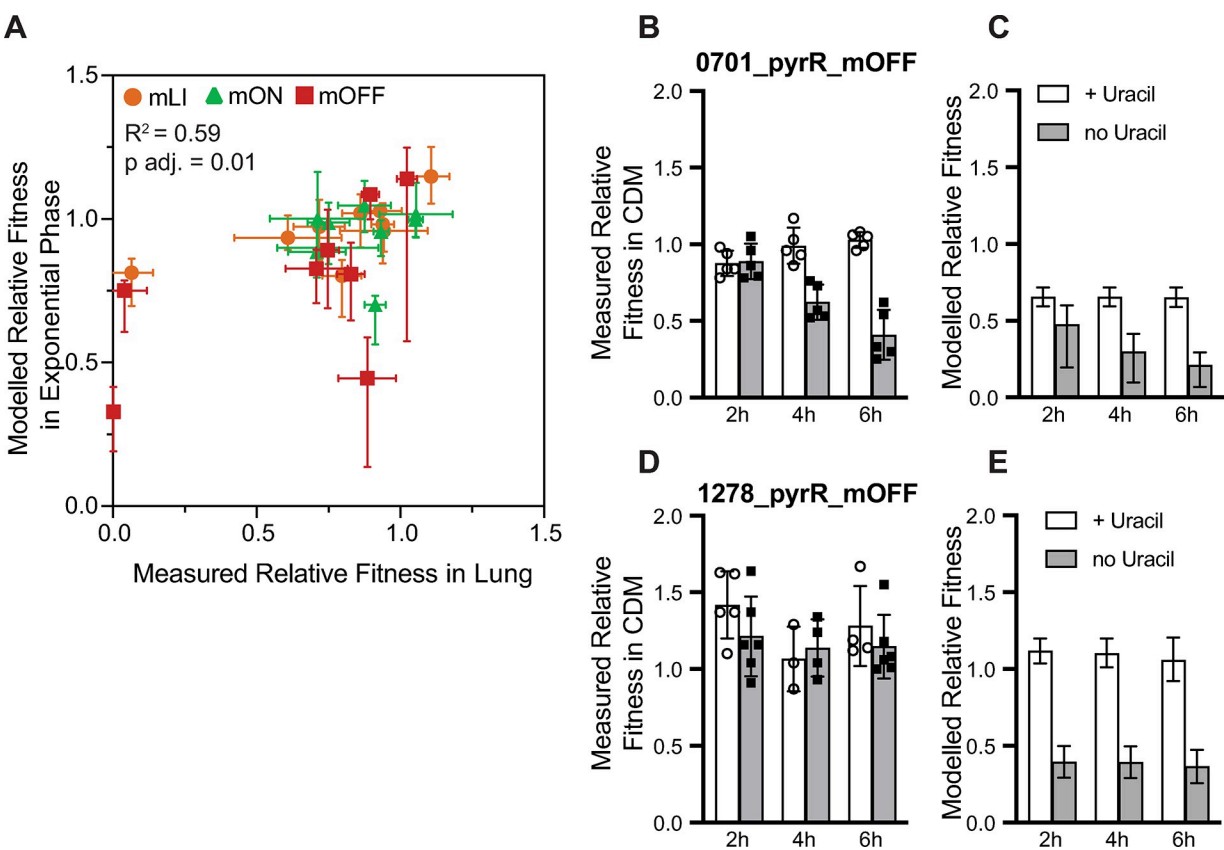

**Fig 5. Modelling strain fitness in co-culture based on monoculture parameters shows weak correlation with *in vivo* fitness.** (A) Strongest correlation (Pearson's correlation co-efficient) observed between modelled fitness (in either stationary or exponential phase) and measured fitness in any *in vivo* environment (nasopharynx, lung, or blood) is between modelled exponential phase fitness and measured fitness in the lung (S7 Fig includes all comparisons). Error bars for measured fitness correspond to standard deviation, error bars on modelled fitness as described in the methods. (B) Measured fitness of strains growth in CDM +/- uracil for strain 0701_pyrR_mOFF compared to the reference strain *S. pneumoniae* TIGR4 at 2, 4, and 6 hours post inoculation. (C) Modelled fitness from for 0701_pyrR_mOFF compared to the reference strain under the same conditions. (D) Measured fitness in CDM +/- uracil for strain 1278_pyrR_mOFF compared to the reference strain *S. pneumoniae* TIGR4 at 2, 4, and 6 hours post inoculation. (E) Modelled fitness modelling for 1278_pyrR_mOFF compared to the reference strain under the same conditions. Similar control experiments with 0701_pyrR_WT[C] and 1278_pyrR_WT[C] are in S6D Fig. All numeric data points in S1 Data.

Thus, in accordance with our initial qualitative observations, we observe only a weak correlation between modelled fitness during exponential phase *in vitro* (equivalent to relative growth rate) and *in vivo* fitness (Figs 5A and S7). Our experimental follow-up further suggests that this weak correlation may be partially due to the impact of co-culture and cross-feeding on parameters measured in monoculture. However, there are also numerous additional factors that influence survival and growth *in vivo* including the dynamic environment and specific virulence processes that have little impact during planktonic growth, but likely contribute to the differences observed between *in vivo* fitness and *in vitro* growth.

## *S. pneumoniae* is sensitive to repression of FMN biosynthesis and transport

While we observe general effects of mis-regulation in our data, and defects in the *in vitro* and *in vivo* environments tested are not strongly correlated, it is also clear that fitness is more sensitive to mis-regulation of some metabolic pathways compared to others. In particular, all our severely deficient phenotypes ($W < 0.25$, relative growth rate or carrying capacity $< 0.5$) are

clustered to regulators associated with two metabolites: FMN and uracil (Fig S4). In each of these cases, growth defects are displayed by mutants with constitutive repression (mOFF) of the biosynthetic pathway in CDM lacking the nutrient (riboflavin or uracil), and one or multiple mutants show a severe defect *in vivo*.

In *S. pneumoniae* there are two apparent routes to obtain FMN: biosynthesis by proteins encoded by the *ribD* operon (SP_0178-SP_0175), or via import of riboflavin from the environment, which is accomplished by the riboflavin transporter encoded by SP_0488 (Fig 6A). Together, these are the only two routes for FMN acquisition in *S. pneumoniae* TIGR4, and the mOFF mutants of both these RNA regulators (0178_FMN_mOFF and 0488_FMN_mOFF) show severe defects, but in different environments. We find that constitutive repression of the *ribD* operon (0178_FMN_mOFF) results in a severe defect in carrying capacity (relative carrying capacity ~0.20) when the strain is grown in CDM lacking riboflavin, but no significant change in doubling time (Fig 6B and S3 Table). This defect is rescued by addition of riboflavin to the medium suggesting a nutritional deficit (Fig 6B). In our reporter gene assays, the 0178_FMN_mOFF mutant displays an approximately 10-fold reduction in β-galactosidase expression compared to the WT repressed condition (S2E Fig), suggesting that our mutant has substantially impaired gene expression of this biosynthetic gene cluster.

To assess whether the constitutively repressed strain is functionally distinct from a complete gene-knockout, we created a strain where the promoter, regulatory region and first gene of the operon (SP_0178, *ribD*) were replaced with a *kan^r* cassette. We found that this strain (SP_0178_KO) behaves similarly to 0178_FMN_mOFF (no statistical differences in growth rate or carrying capacity between the strains grown under the same conditions, S3 Table) suggesting that the mOFF constitutive repression behaves similarly to a gene knockout (Fig 6D). However, one aspect of the growth curves not captured by our quantitative analysis is the autolysis of *S. pneumoniae*, which results in a decline in culture density ($OD_{600}$) following exponential growth. Autolysis is characteristic for *S. pneumoniae* TIGR4 grown in rich medium (S3M Fig), but often not observed in CDM (S3 and S6D Figs). The SP_0178_KO strain appears to undergo autolysis in CDM (Fig 6D) whereas our other strains typically do not, or do so after a more extended plateau. This decline may indicate that this strain is under some additional stress that is not captured by our other measures.

The 0178_FMN_mOFF strain also displays a mild defect in nasopharynx colonization ($W = 0.81$), but no apparent defect in lung infection or transition to the blood ($W > 0.9$) (Fig 6E). This result is consistent with previous transposon insertion sequencing findings that the entire riboflavin biosynthesis operon (SP_0178-SP_0175) is dispensable during infection in TIGR4 [76], and suggests that riboflavin is prevalent and *de novo* biosynthesis unnecessary in this environment. In contrast, constitutive repression (mOFF) of the transporter (*ribU*, SP_0488) results in a strongly deleterious phenotype in all *in vivo* environments (S6F Fig). This finding is also consistent with past transposon insertion sequencing experiments showing that SP_0488 is essential for nasopharynx colonization ($W = 0$) [76]. It is surprising that expression of the transporter is essential during infection given an intact biosynthesis pathway. However, based on our β-galactosidase assays utilizing the native promoters and regulatory regions, the transporter (SP_0488) is expressed approximately 10-fold higher than the biosynthesis operon under both conditions (CDM + and–riboflavin), with the repressed expression of the transporter (~80 Miller units) exceeding the expression of the derepressed biosynthetic operon (~20 Miller units). Our results imply that the synthesis operon may not constitute a significant source of FMN during infection, or that the transporter plays an essential role during infection aside from riboflavin transport.

All three mutants of 0488_FMN also show a mild to moderate defect in growth rate and carrying capacity in planktonic culture (Fig 6C). However, this defect affects all of the mutants,

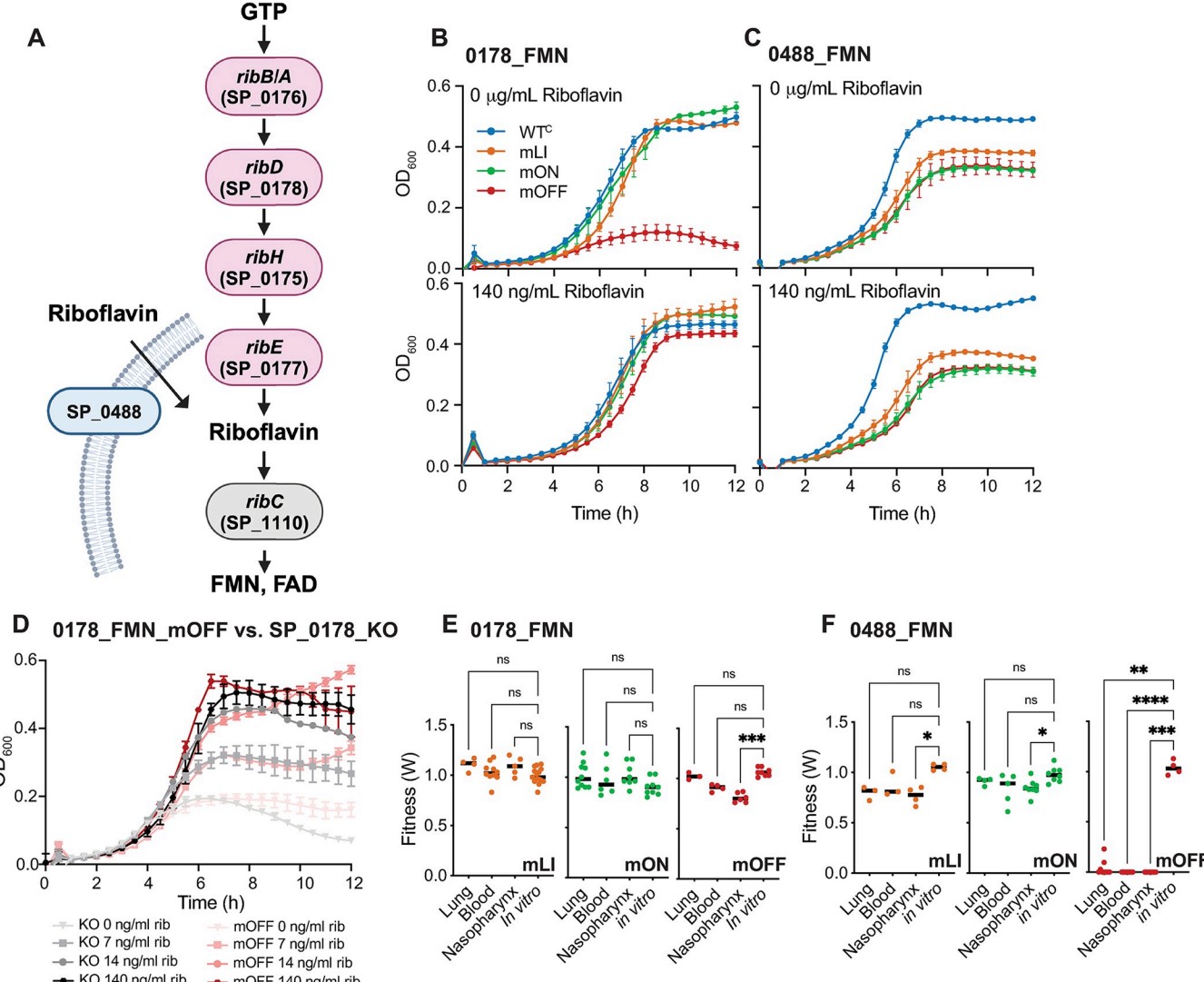

**Fig 6. Constitutive repression of riboflavin biosynthesis and transport is deleterious. (A)** FMN biosynthesis pathway indicating the roles of different riboswitch regulated genes in the synthesis and transport of riboflavin. Pink genes are biosynthesis genes, turquoise genes are transporters, and grey genes are not riboswitch regulated. Created with BioRender. **(B)** Growth curves for 0178_FMN WT[C] and mutants in CDM +/- riboflavin shows that 0178_FMN_mOFF displays a severe growth defect in medium lacking riboflavin. **(C).** Growth curves for 0488_FMN WT[C] and mutants in CDM +/- riboflavin shows that all three 0488_FMN mutants displays a moderate defect that is riboflavin independent. **(D)** The SP_0178_KO strain where the promoter, regulatory region and coding region of SP_0178 have been removed shows no significant change in growth rate and carrying capacity compared to the constitutively repressed mutant 0178_FMN_mOFF at a range of riboflavin concentrations (S3 Table). **(E)** Mouse competition assays demonstrate that 0178_FMN_mOFF mutants display a statistically significant fitness defect in nasopharynx colonization, but no other environments. 0178_FMN_mLI and 0178_FMN_mON mutants do not show any significant *in vivo* defects. Each point represents fitness calculated from an individual mouse. **(F)** Mouse competition assays demonstrate that 0488_FMN_mOFF mutants display a statistically significant fitness defect in all *in vivo* environments. 0488_FMN_mLI and 0178_FMN_mON mutants only display a statistically significant defect in the nasopharynx. Each point represents fitness calculated from an individual mouse. Statistical significance for mouse studies assessed via Kruskal-Wallis test followed by Dunn's test of multiple comparisons to compare each *in vivo* environment to the *in vitro* control (*adj. p<0.05, ** adj. p < .01, ***adj. p < .001). All numeric data points in S1 Data.

and appears to be riboflavin independent (Fig 6C). Since the WT[C] strain behaves similarly to an unmodified strain, this defect does not seem to be attributable to the position of the Chl[r] marker. Furthermore, the constitutively active 0488_FMN_mLI and 0488_FMN_mON strains only display mild defects in our animal studies ($W$ = 0.76 and 0.85 in the nasopharynx, Fig 6F), suggesting that the source of this growth defect *in vitro* is not directly related to the likely

nutrient deficiency impacting the constitutively repressed 0488_FMN_mOFF in the context of colonization and infection.

## *S. pneumoniae* is sensitive to mis-regulation of pyrimidine synthesis

We also observed exceptional sensitivity of *S. pneumoniae* to perturbation of pyrimidine synthesis. In *S. pneumoniae* pyrimidine synthesis and transport is highly regulated. Well-conserved pyrR RNA regulators that interact with a UMP-bound PyrR protein to inhibit gene expression precede operons containing biosynthesis genes (SP_0701, SP_0702 (*pyrE/F*), SP_1277-SP_1275 (*pyrB*, *carA*, *carB*)), and a putative uracil transporter (SP_1286) (Fig 7A and 7B). In addition to its potential role in pyrimidine salvage [81], PyrR (SP_1278) is the regulatory protein, and thus may have trans-acting effects on other portions of pyrimidine synthesis and transport. Similarly to the FMN riboswitches, the combined action of the pyrR regulators in *S. pneumoniae* allows tight regulation of both uridine synthesis and transport with no obvious alternative sources.

We observe that constitutive repression of pyrimidine biosynthesis genes (0701_pyrR_mOFF, and 1278_pyrR_mOFF) results in significant growth defects in culture lacking uracil (Figs 7C and S3L, [64]). In both cases, relative growth rate and relative carrying capacity are substantially reduced in this condition. For 1278_pyrR_mOFF, this defect is completely rescued by the addition of uracil to the medium, but for SP_0701_mOFF, the growth rate defect remains when the medium is supplemented with uracil (relative growth rate = 0.71 of WT$^C$). To investigate whether addition of more uracil to the medium would completely rescue the phenotype, we increased uracil concentration in the medium up to five times its standard concentration to 100 μg/mL. We found that the higher concentrations of uracil beyond 20 μg/mL have no discernable impact on the growth of the mOFF mutant (Fig 7E and S3 Table), suggesting that the cost of constitutively repressing this portion of pyrimidine biosynthesis goes beyond uracil insufficiency.

To assess how our repressed gene expression phenotypes observed compared to complete operon knockouts, we created *S. pneumoniae* TIGR4 strains where the entire operon was removed and replaced with a *kan*$^R$ cassette for SP_0701 (deletion of SP_0701 *pyrF* and SP_0702 *pyrE*) and SP_1278 (deletion of SP_1278 *pyrR*, SP_1277 *pyrB*, SP_1276 *carA*, and SP_1275_*carB*). For both of these strains we find that the knockouts require higher levels of nutrient supplementation to rescue the phenotypes compared with our constitutive repression strains (0701_pyr_mOFF and 1278_pyr_mOFF) (Fig 7F and 7G and S3 Table). These findings further support that our phenotypes in culture are likely the result of insufficient gene product, but are not necessarily as severe as complete operon knockouts.

Assessing the behavior of the constitutively repressed pyrR mutants *in vivo* shows that as 1278_pyrR_mOFF is severely deleterious in all *in vivo* environments (previously published [64]), but that 0701_pyrR_mOFF shows no significant phenotype under any *in vivo* environment (Fig 7H). These findings are consistent with previous results from transposon insertion sequencing indicating that insertions within SP_1278 result in defects to lung infection ($W$ = 0.48) [76]. While there is no information on the impact of insertions within SP_1278 on nasopharynx colonization available, those within SP_1277 (but not SP_1276 or SP_1275) result in defects in nasopharynx colonization ($W$ = 0) [76]. Insertions in SP_0701 do not display defects *in vivo*. We were unable to generate a regulatory mutant with a constitutively reduced SP_1286 expression for comparison, however, there is also no significant defect associated with insertions in SP_1286 in either environment in transposon insertion sequencing studies [76]. The combination of these data suggests that reduction in levels of the regulatory protein encoded by SP_1278 may be especially deleterious in the *in vivo* environment, while just disrupting pyrimidine synthesis may not be so problematic.

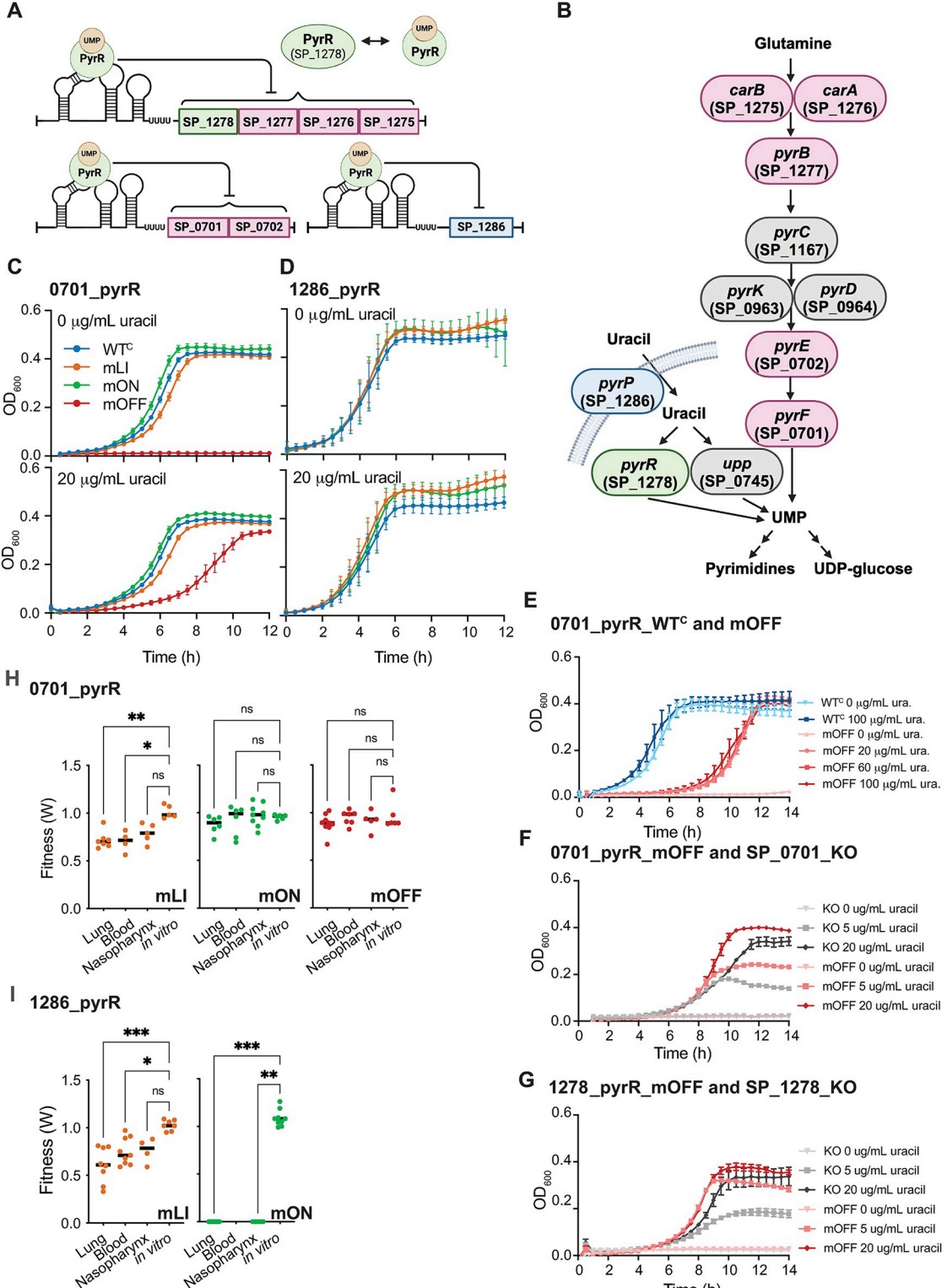

**Fig 7. Pyrimidine synthesis and transport is very sensitive to mis-regulation. (A)** PyrR regulated operons are repressed by the PyrR protein (encoded by SP_1278) bound to UMP. Figure created with Biorender. **(B)** Pyrimidine synthesis pathway indicating the roles of pyrR RNA regulated genes in the synthesis and transport of pyrimidines. Pink genes are biosynthesis genes, turquoise genes are transporters, and grey genes are not pyrR regulated. PyrR (SP_1278) may show both regulatory and enzymatic activity. Figure created with Biorender. **(C)** Growth curves for 0701_pyrR WT[C] and mutants in CDM +/- uracil shows that 0701_pyrR

_mOFF displays a severe growth defect in medium lacking uracil that is not fully rescued in medium containing uracil. **(D).** Growth curves for 1286_pyrR WT$^C$ and mutants in CDM +/- uracil shows no significant growth defects. Note, that no mOFF constitutively repressed mutant was analyzed for 1286_pyrR. **(E)** Adding additional uracil to the medium (from 20 ug/mL to 100 ug/mL) does not alter the uracil independent growth defect observed for SP_0701_mOFF (S3 Table). **(F)** The SP_0701_KO strain where the promoter, regulatory region and coding region of SP_0701 and SP_0702 have been removed shows a more severe phenotype compared to the constitutively repressed mutant 0701_pyrR_mOFF at a range of uracil concentrations (S3 Table). **(G)** The SP_1278_KO strain where the promoter, regulatory region and regions inclusive of SP_1278 through SP_1275 were removed shows a more severe phenotype compared to the constitutively repressed mutant SP_1278_pyrR_mOFF at a range of uracil concentrations (S3 Table). **(H)** Mouse competition assays demonstrate that 0701_pyrR _mLI mutants display a statistically significant fitness defect in lung infection and the associated transition to blood. 0701_pyrR_mON and 0701_pyrR_mOFF do not show significantly decreased fitness in mouse infections. Each point represents fitness calculated from an individual mouse. **(I)** Mouse competition assays demonstrate that 1286_pyrR _mLI and mON mutants display a statistically significant fitness defect in most infection environments. The 1286_pyrR_mOFF mutant was not successfully constructed, and therefore could not be tested (Fig 1G). For 1286_pyrR_mON, no colonies were recovered from the blood despite repeated attempts strongly suggesting a fitness very close to 0, but that is ultimately not quantifiable. Each point represents fitness calculated from an individual mouse. Statistical significance of *in vivo* defects for mouse studies assessed via Kruskal-Wallis test followed by Dunn's test of multiple comparisons to compare each environment to the in vitro rich medium control (*adj. p<0.05, ** adj. p < .01, ***adj. p < .001). Data corresponding to growth curves and mouse infections for mutants of SP_1278 is previously published [64]. All numeric data points in S1 Data.

In contrast to our culture conditions where over-expression causes no significant defects (Fig 7C and 7D), we find that constitutively active pyrR regulators have deleterious impacts on fitness *in vivo*. The mLI mutations display deleterious impacts on *in vivo* fitness that vary from severe in the case of 1278_pyrR_mLI ($W$ <0.1 in all *in vivo* environments [64]) to milder, but still statistically significant, for 0701_pyrR_mLI and 1286_pyrR_mLI ($W$ = 0.6–0.75 in the lung infection and blood) (S4 Table and Fig 7H and 7I, [64]). The mON mutations with constitutive activity have more varied phenotypes from severely deleterious in the case of 1286_pyrR_mON ($W$ < 0.1 in the blood and nasopharynx environments), to no significant impact in the case of 0701_pyr_mON. Differences between these phenotypes and those displayed by the mLI mutants may be due to differences in the final level of constitutive expression for each mutant (S2J and S2K Fig, [64]), and point toward a situation where the absolute gene expression levels during infection may be quite critical for fitness. Thus, pyrimidine biosynthesis and transport appear to be exceptionally sensitive to changes in gene expression with both constitutive expression and constitutive repression resulting in severe defects during infection. Furthermore, these defects seem less tied to nutrient deficiency, and more to other aspects of survival or growth in the infection environment.

## Discussion

In this work we validated the regulatory activity, and constructed three mutant strains designed to have constitutively active (mON), repressed (mOFF) gene expression, or ligand insensitivity (mLI), for a total of twelve RNA cis-regulatory elements in *Streptococcus pneumoniae* TIGR4. By assessing the fitness of these strains in both chemically defined medium (CDM) as well as three infection environments: nasopharynx colonization, lung infection, and transition to the blood upon lung infection (blood), we demonstrate that constitutively active and repressed regulatory mutants are significantly less fit than matched controls in most *in vivo* environments, even as this difference is not apparent for mutants with constitutive gene expression in planktonic culture (Fig 4).

Aggregated across the entire set of regulators, our data suggests that the selection pressure to maintain regulation when measured in environments mimicking a natural environment of the organism is measurable with the median relative fitness across all our mutant strains, including both constitutively active and repressed phenotypes, approximating $W$ = 0.85 (mean $W$ across all variants is = 0.75). Thus, a conservative estimate of the selection coefficient is ~0.15 (where $s$ = 1-$W$). The *in vivo* fitness deficits observed for many individual regulators in

this study may seem modest (W > 0.75). However, in organisms with large populations such as *S. pneumoniae*, even a modest deficit is more than sufficient to drive evolutionary dynamics. Work in understanding the rise of antibiotic resistance suggests that selection coefficients in the range of 0.01 to 0.1 easily enrich a population for resistant variants in a few hundred generations [82]. Thus, these selection coefficients are more than sufficient to drive the conservation and maintenance of these regulatory sequences in the *S. pneumoniae* core genome.

That both constitutive repression (mOFF) and constitutive activation (mON) lead to decreases in fitness may be considered somewhat paradoxical. However, *in vivo* fitness is a property that encompasses many different characteristics including survival in an adverse environment, successful attachment to and invasion of host cells, and ultimately efficiency of replication. As such, under- and over-expression may impact different characteristics, yet ultimately yield similar impacts on fitness.

Our work also revealed that specific pathways appear to be more sensitive to constitutive repression or mis-regulation. In the case of FMN, it appears that either biosynthesis or transport is critical for fitness depending on the environment. Surprisingly, the riboflavin biosynthetic operon appears to be largely dispensable *in vivo*, and cannot compensate for lack of riboflavin transport. This finding has important consequences for endeavors to target FMN riboswitches with antimicrobials [38,39,83–85]. This is an effective pathway to target with antimicrobials. Mutations resulting in lack of riboflavin, which would be mimicked by the drug action, result in severely deleterious phenotypes both in culture and *in vivo*. However, resistance mutants derived in culture to compounds targeting the FMN riboswitch arise relatively easily through mutation of the aptamer, typically resulting in constitutive expression of riboflavin biosynthesis [38,39]. Given the lack of deleterious consequences for such mutations *in vivo*, it is likely that such mutations may also be problematic in the clinic.

In the case of pyrimidine synthesis and transport, both constitutive activation and repression may be severely deleterious *in vivo*, while only constitutive repression is deleterious in planktonic culture (Fig 7). The 1278_pyrR regulator appears to be the most sensitive, showing the most universal deleterious phenotypes *in vivo*. We speculate that this sensitivity is due to the role of the PyrR regulatory protein. Altering levels of the regulatory protein likely results in trans-acting effects on other sites [72]. This speculation is supported by previous work showing that *S. pneumoniae* ΔSP_2193 (two-component system histidine kinase), resulting in repression of pyrimidine synthesis genes, suppresses fitness defects resulting from deletion of an individual pyrimidine biosynthesis genes. Collectively these findings suggest a tight link between pyrimidine synthesis and processes that are essential for infection.

One potential connection between pyrimidine levels and virulence is suggested by work in *S. pneumoniae* D39 showing connections between uracil levels and capsule production [86,87]. This finding is supported by the fitness defects observed for strains with transposon insertions into pyrimidine biosynthesis genes (SP_0701, SP_0702, SP_1277, SP_1276, SP_1275) when grown in culture medium with various carbohydrate sources including: glucose, sucrose, cellobiose, raffinose, maltose, mannose, and GlcNac [76]. Thus, the ultimate impact of pyrimidine regulation during infection may be on sugar processing via UDP-glucose for capsule formation rather than solely pyrimidine availability for growth. However, pyrimidine biosynthesis has been shown to be essential for *in vivo* growth in several other bacterial pathogens, so the impact of inhibiting pyrimidine synthesis may be more widespread and not necessarily specific to *S. pneumoniae* [88,89].

While few of the regulators outside these two pathways display severe phenotypes, they do systematically display mild fitness defects. In contrast to both the FMN and pyrimidine pathways, none of the other regulators assessed here have a clear route for controlling nutrient access *in toto*. Both the glycine riboswitch (0408_glycine) and tryptophan T-box (1069_TrpT-

box) regulate transporters that are responsible for importing the amino acid into the cell (S1 Fig). In *S. pneumoniae* TIGR4 there are biosynthesis pathways for these nutrients that are not similarly regulated. Glycine can be synthesized directly from serine (*glyA*, SP_1024), and there is an intact pathway for tryptophan synthesis from chorismate (*trpA-E, G*, SP_1811-SP_1816). Similarly, the guanine riboswitch (1847_guanine) in *S. pneumoniae* TIGR4 regulates *xpt* (SP_1847, xanthine phosphoribosyltransferase) and *pbuX* (SP_1848, xanthine permease), which are responsible for purine biosynthesis and import, respectively. However, *xpt* is functionally redundant with a distally located gene, *htp* (hypoxanthine-guanine phosphoribosyltransferase, SP_0012). Thus, in all three cases, the RNA regulators are most closely associated with transport, and the biosynthesis is able to proceed outside such regulation. Finally, there are four instances of the TPP riboswitch that regulate a host of different transporters and thiamine pyrophosphate biosynthesis genes (0716_TPP, 0719_TPP, 0726_TPP, and 2199_TPP), not all of which are well-characterized. Several of these regulatory units are also somewhat redundant with each other (e.g. *tenA*, *thiE* and *thiM* are all present in 2 copies, S1 Fig). Across all the regulators examined, it is likely that such redundancies blunt the impact of mis-regulation by our individual regulatory mutants.

In addition, while we have assayed the gene-expression of our mutants in culture, we do not have direct evidence of the gene-expression during infection. In particular, the regulatory mechanisms we assess here are all post transcription initiation, thus the genes must be transcribed for mutations to have a functional impact. Our findings suggest that the transcripts are produced *in vivo*, as at least one mutant of each regulator displayed a statistically significant fitness deficit in one or more *in vivo* conditions. In addition, assessment of existing dual RNA-seq data from TIGR4 in these environments indicates that the studied transcripts are detectable during infection [90], although the extent of expression varies widely. Notably, SP_0488, whose expression we found to be critical for viability during infection is the most highly expressed of these transcripts *in vivo* (S6 Table). Thus, we anticipate that our mutants with constitutive activation or repression likely behave similarly *in vivo* as they do *in vitro* with similarly increased or decreased expression. However, the caveat that there may be additional regulatory mechanisms acting that are not apparent *in vitro* remains.

In conclusion, we measured the fitness advantage conferred on *S. pneumoniae* by a series of RNA-regulatory mutants to find that fitness is significantly decreased within *in vivo* environments across both constitutively active and repressed gene expression mutants. The lack of correspondence between these values with behavior during *in vitro* planktonic culture indicates that in evaluating the impact of regulation on organismal fitness assessment of *in vivo* environments is necessary for appreciating the biological role of such mechanisms. The differences observed between *in vitro* and *in vivo* environments may be due to a number of factors including the impact of a co-culture vs. monoculture assay, the dynamic environment encountered by populations growing *in vivo*, or *in vivo* nutrient limitations not well mimicked by standard culture conditions. Although often considered fine-tuning mechanisms [75], cis-regulatory RNA features play key roles in mediating *in vivo* success.

## Methods

### Analysis of RNA regulator sequences

Genomes of 33 representative *S. pneumoniae* strains were downloaded from NCBI (BioProject: PRJNA514780). Each genome was scanned with cmscan [91] with lowered thresholds to ensure detection of RNA cis-regulators. Complete sequences were extracted from each genome, and re-aligned and compared via clustalOmega [92]. Downstream protein coding genes were identified in the same set of genomes via nucleotide similarity (BLAST) with

regions in *S. pneumoniae* TIGR4, and compared in a similar manner. Sequence accession numbers for protein and RNA nucleotide sequences compared are listed in S1 Table.

## B-Galactosidase reporter strain construction and activity assay

To create reporter strains of *S. pneumoniae* TIGR4 we replaced the native membrane bound β-galactosidase (SP_0648) with a kanamycin resistance cassette (kan[r]) and a β-galactosidase reporter sequence (*lacZ*) that lacked a promoter [93]. Subsequently, the kan[r] marker was replaced with a chloramphenicol marker (Chl[r]) accompanied by the regulatory region including each RNA (or their mutant) translationally fused to *lacZ* expressing a soluble β-galactosidase protein [93]. Primers used to generate the regulatory RNA mutant reporter strains are listed in S7 Table. To perform β-galactosidase activity assays (Miller Assays), *S. pneumoniae* regulatory RNA mutant *lacZ* reporter strains were cultured to mid-log phase in chemically defined media (CDM) (S2 Table) in the presence or absence of the respective ligand. Cells were resuspended in Z buffer (50 mM $Na_2HPO_4$, 40 mM $NaH_2PO_4$, 10 mM KCl, 1 mM $MgSO_4$, 50 mM 2-mercaptoethanol) and β-galactosidase activity assay was performed as previously described with the substrate ONPG [64,94]. For statistical analysis of individual WT strains, an unpaired t-test was conducted in Prism (Graphpad version 9.5.1). For analysis of mutant strains, one-way ANOVA followed by Sidak's multiple comparisons test were conducted for each set of mutants in Prism (Graphpad version 9.5.1).

## Expression analysis using qRT-PCR

Total RNA was isolated from cultures grown as described above for β-galactosidase assays using the Qiagen rNeasy kit and treated with TURBO DNase (ThermoFisher). DNA-free RNA was used to generate cDNA using iScript reverse transcriptase supermix (BioRad), which was used to perform the quantitative PCR using QuantStudio (ThermoFisher). Each sample was normalized against the L33 50S ribosomal protein gene (SP_0973). Samples were measured in biological duplicate and technical triplicate, including no-reverse transcriptase and no-template controls. Efficiency of primers used for qRT-PCR was validated using dilutions of genomic DNA and the primer sequences are listed on S7 Table. For statistical analysis an unpaired t-test was implemented in Prism (Graphpad version 9.5.1). For analysis of mutant strains, one-way ANOVA followed by Sidak's multiple comparisons test were conducted for each set of mutants in Prism (Graphpad version 9.5.1).

## Regulatory RNA mutant strain construction

Regulatory RNA mutants were generated as previously described [64]. Briefly, each wildtype regulatory RNA element was amplified along with ~1 kb of homologous flanking regions from *S. pneumoniae* TIGR4 genomic DNA (GeneBank accession number NC_003028.3). Mutations to the regulatory RNA were created using appropriate primers (S7 Table) and were assembled along with the chloramphenicol resistance cassette using Gibson assembly. The amplified, assembled product was used to transform *S. pneumoniae* such that the mutant copy of the RNA regulator replaces the native copy in the genome. Transformants were screened for chloramphenicol resistance and mutations confirmed by Sanger sequencing of an appropriate amplicon from genomic DNA of the mutant strain.

## RNA regulator mutant growth assay

The wild type strain with the chloramphenicol cassette (WT[C]) and regulatory RNA mutants of each riboswitch were cultured in a complete defined media (CDM, S2 Table) lacking their

respective ligand. Optical density (OD) was adjusted to 0.015 and the cells were cultured in CDM with and without the ligand or ligand precursor in flat-bottom 96-well plates. Ligands were present at the following concentrations 0.3 μg/mL thiamine, 140 ng/mL riboflavin, 20 μg/mL uracil, 75 μg/mL glycine, and 204 μg/mL tryptophan. Only guanine is not a standard component of CDM, in this case CDM was supplemented with 50 μg/mL guanine in the ligand present condition. $OD_{600}$ measurements were taken on a BioSpa 8 plate reader (Agilent) at 30-minute intervals. Experiments were done with technical duplicates or triplicates, and in biological duplicate or triplicate.

Growth curves were analyzed using the R package growthcurver [77] to extract the maximum growth rate (r) and carrying capacity (K) from each individual curve. Time points between 1 and 12.5 hours were used for most analyses. Statistical significance between the values of each mutant and the $WT^C$ was determined by comparing parameter values (doubling time = ln(2)/r, and K) from individual growth curves via a Kruskal-Wallis test followed by Dunn's test of multiple comparisons (Graphpad version 9.5.1). Growth rates and carrying capacity values were subsequently normalized to the average value for the $WT^C$ strain within a specific culture condition (e.g. CDM -riboflavin) for presentation in S4 Fig. The values on S3 Table represent the averages and standard deviations of parameters across all growth curves, typically 5–9 replicates in 2 or 3 biological replicates with technical duplicate or triplicate.

To assess growth rates for select mutants, medium conditions were altered as described in the main text and figure legends. Specifically, CDM was supplemented with additional uracil (20 μg/mL, 40 μg/mL, and 100 μg/mL), uracil was titrated into CDM (0 μg/mL, 5 μg/mL, and 20 μg/mL), and riboflavin was titrated into CDM (0 ng/mL, 7 ng/mL, 14 ng/mL, 140 ng/mL). Datasets for 0178_FMN_KO were truncated (analysis of 1–6 hours) to prevent decline associated from autolysis from impacting curve fit. Statistical difference between pairs of strains under a specific condition (e.g. KO vs mOFF mutants) assessed via a Mann-Whitney test in Prism (Graphpad version 9.5.1).

### *In vivo* regulator RNA mutant fitness determination

Competition assays in mouse infection model were performed essentially as previously described [64]. Briefly, regulatory RNA mutants (mLI, mON, mOFF) and their respective $WT^C$ were cultured separately in rich media (THY or semi-defined minimal media-SDMM) mixed with the *S. pneumoniae* TIGR4 wild type strain in approximately a 1:1 ratio. Using this mixture mice were infected intranasally with ~0.5–1.5 x $10^7$ total CFU in 40 μL for lung infection, or ~5 x $10^7$ CFU in 10 μL for nasopharynx colonization. A control competition was conducted by inoculating 5 μL of this mixture into 5 mL of a semi-defined rich medium (SDMM) and allowing growth until $OD_{600}$ = ~0.5.

For lung infections, mice were sacrificed 20–24 hours post infection and the lungs removed and homogenized in 1 ml PBS, and blood collected (up to 500 μl). For nasopharynx colonization, nasopharyngeal lavage (500 μl) was collected at 24–48 hours post infection. In instances where mutants were cleared within 24 hours (1286_pyrR_mON), samples were collected at 6–8 hours post infection in attempts to obtain a fitness measurement. Experiments involving animals were performed with approved protocols from the Boston College Institutional Animal Care and Use Committee (IACUC) (protocol #'s: 2019–009, 2022-011-01).

Serial dilutions of recovered samples were plated onto blood plates (-/+ chloramphenicol) to enumerate the total number of CFU, and CFU originating from the RNA mutant ($Chl^r$) in the pre-infection mixture and in each sample. As in past experiments [95], from these enumerations, the ratio of mutant/total CFU was determined for each condition and used to calculate

the fitness ($W$) as:

$$W = \frac{\ln[F_{t2}*d/F_{t1}]}{\ln[(1 - F_{t2})*d/(1 - F_{t1})]}$$

in which $F_{t1}$ and $F_{t2}$ are the frequency of the mutant (number of Chl$^r$ CFU/total CFU) at the start $t_1$ and the end $t_2$ of the experiment, and $d$ (expansion factor) accounts for growth in the total population [78]. For *in vitro* competitions, this value is based off the starting and ending OD$_{600}$. For mouse studies, doubling times corresponding to 108 minutes for the lung and 161 minutes for the nasopharynx [76] and result in $d$ = 10321 ($2^{13.3}$) and $d$ = 492 ($2^{8.944}$) for 24 hour infections of the lung and nasopharynx respectively. These values were adjusted to reflect the timing of each infection as necessary. To assess significance of each individual regulator mutant, values across environments were compared to those obtained from the *in vitro* rich medium control using Kruskal-Wallis test followed by Dunn's test of multiple comparisons in Prism (Graphpad version 9.5.1).

## Statistical assessment of aggregated data

To aggregate data across regulators the relative growth rate and relative carrying capacity (relative to the mean WT$^C$ value for each regulator) under each condition were randomly sampled without replacement to obtain n = 3 replicates for each mutant of each regulator. One hundred random samplings were performed via the R function *sample* (R version 4.3.1). For each parameter (relative growth rate and relative carrying capacity) under each condition (+ and– target ligand) the sampled populations of mutants were compared to sampled WT$^C$ values using Kruskal-Wallis test (*kruskal.test* in R stats package) followed by Dunn's test of multiple comparisons (*dunntest* from R FSA package [96]) to determine significance (S5 Table). A similar analysis was performed for the fitness values obtained for each strain under each tested condition (*in vitro* (rich medium), nasopharynx, lung, and blood) (S5 Table).

## Modelling of strain growth and relative fitness

To model the expected fitness in a co-culture experiment, the following system of differential equations was solved with the ODE45 numerical solver in Matlab (Mathworks).

$$\frac{dN_a}{dt} = r_a \frac{N_a}{K_a} \left( max \left( 0 | K_a - N_a - N_b \right) \right)$$

$$\frac{dN_b}{dt} = r_b \frac{N_b}{K_b} \left( max \left( 0 | K_b - N_b - N_a \right) \right)$$

where $r_a$ is the growth rate of strain A, and $r_b$ the growth rate of strain B, $K_a$ the carrying capacity of strain A, $K_b$, the carrying capacity of strain B. From the modeled populations, the fitness was calculated as above based on the ratios of strain B to the total population (strain A + strain B) at each time point. The default growth rates used for the reference strain for initial modeling are $r_a$ = 0.0252 min$^{-1}$, and $K_a$ = 0.462, based on the average of measured values for WT$^C$ strains in CDM (including all components at standard concentrations) over all experiments. The values for $r_b$ and $K_b$ for calculations reported in S6B and S6C Fig were determined by scaling the default values of $r_a$ and $K_a$ by the relative growth rate and carrying capacity values reported on S3 Table (i.e. $r_b = r_a$*(relative growth rate for mutant)). The starting values of $N_a$ and $N_b$ were set to 0.001. Our calculation of fitness is generally not sensitive to the starting ratio of $N_a/N_b$, and ratios in actual experiments generally did not differ substantially from 1:1. Error was estimated by repeating calculations with values of $r_a$ and $K_a$ adjusted by the standard deviations

reported on S3 Table. Exponential phase fitness taken at 100 minutes, and stationary phase at 500 minutes. The values reported on S6B and S6C Fig reflect those calculated from parameters collected in CDM lacking the target nutrient.

To estimate *W* for *in vitro* competition in CDM +/- uracil, $K_a$ was adjusted to 1.1 to account for the difference in pathlength and volume for larger culture tubes used for these assays. For such calculations, the reference strain growth rate and $N_a$ were used as above, but measured growth rates under each condition were used directly rather than adjusted based on the $WT^C$ strain reading (e.g. $r_b$ = growth rate on S3 Table), $K_b$ adjusted from 1.1 based on the relative carrying capacity of the strain, and $N_b$ as above. *W* calculated at 120, 240, and 360 minutes.

### *In vitro* mutant fitness determination under selective conditions

Regulatory RNA mutants along with their respective $WT^c$ were pre-grown in a rich media (SDMM) to mid-log phase. Mutant cells were centrifuged, washed with PBS, and mixed in a 1:1 ratio with the *S. pneumoniae* TIGR4 316 cultured separately. Serial dilutions of the mixture were plated as above to accurately enumerate the starting ratio. The mixture was also used to start 5 ml CDM culture with or without respective ligand in technical duplicate or triplicate and biological duplicate. At each time point (2, 4, 6 hours), $OD_{600}$ was determined and 10ul of the culture was serially diluted and plated to determine CFU/ml. Fitness was calculated as above.

### Supporting information

**S1 Table. Accession numbers for the regulatory RNA and downstream protein used to assess extent of conservation across representative *S. pneumoniae* genomes.**
(XLSX)

**S2 Table. Composition of the Complete Defined Medium (CDM) used to assess extent of gene expression and growth of mutant strains.** Medium is made, filter sterilized and used within 3 days of creation.
(XLSX)

**S3 Table. Measured parameters from growth curves in S3 Fig (doubling time, growth rate, carrying capacity, relative growth rate, and relative carrying capacity) for each individual mutant under +ligand (CDM with no dropouts for all RNAs except 1847_guanine, which was guanine supplemented as noted) and -ligand (CDM lacking specific ligand or ligand precursor as noted) conditions.** Statistical significance of difference between each mutant to the $WT^C$ is noted. Growth parameters derived from curves in Figs 6D, 7D and 7E.
(XLSX)

**S4 Table. Average relative fitness values for each RNA regulator during nasopharynx colonization, lung infection, and transmission to the blood.** Statistical significance between each the *in vivo* environments and an *in vitro* control noted for each individual strain tested is noted. Individual points are in S5 Fig.
(XLSX)

**S5 Table. Average p-values and proportion of samples with p-values <0.05 from data aggregation and sampling analysis.**
(XLSX)

**S6 Table. Raw counts and transcripts per million for genes regulated by RNA cis-regulators studied in mice during nasopharynx colonization, lung infection, transmission to the**

blood [90].
(XLSX)

**S7 Table. Primers used for strain construction and qPCR.**
(XLSX)

**S1 Fig. Genes regulated by each RNA cis-regulator chosen for study in *S. pneumoniae*
TIGR4 categorized by the type of regulator.** Locus tags and gene names provided where
available. Gene function (transport, biosynthesis, regulatory, unknown) indicated by the color
of the box. Operon structure determined in a previous sequencing study [64]. #This regulator
was assessed as part of a previously published study [64].
(PDF)

**S2 Fig. Regulatory RNA putative secondary structures and mutant gene expression.** (**A-K**):
Putative secondary structures are derived from aptamer consensus folding [65] and minimum
free energy calculations [97]. Mutations are indicated on each structure. Some mutations are
large deletions where the deleted nucleotides are bracketed by an appropriate color bracket
(Ligand Insensitive = yellow, ON = green, OFF = red). β-galactosidase activity measured via
Miller assay [94], and error bars represent standard deviation and individual biological repli-
cates are indicated by points. Significance of mutant changes in activity upon ligand binding
determined via one-way ANOVA followed by Sidak's multiple comparisons test to compare
values in the + and–ligand conditions for each mutant. ($^*p<0.05$, $^{**}p<0.01$, $^{***}p < .001$). WT
samples are duplicated from Fig 1A for reference. (**A-D**) TPP riboswitch structures and
mutants' β -galactosidase activity in CDM lacking thiamine and including thiamine
(-/+ 0.3 μg/ml thiamine). (**E, F**) FMN riboswitch structures and mutants' β -galactosidase
activity in CDM (-/+140 ng/mL riboflavin). (**G**) Glycine riboswitch structure and mutants' β
-galactosidase activity in CDM (-/+75 μg/mL glycine). % indicates values that are significantly
different from one another, but not considered ligand responsive due to the direction of the
response. (**H**) Tryptophan T-box structure and mutants' β -galactosidase activity in CDM (-/+-
204 μg/mL tryptophan). (**I**) Guanine riboswitch structure and mutants' β -galactosidase activi-
ty in CDM and CDM supplemented with 50 ug/mL guanine. (**J**) PyrR element preceding
SP_0701 structure and mutants' β-galactosidase activity in CDM (-/+ 20 μg/mL uracil). (**K**)
PyrR element preceding SP_1286 and qRT-PCR measurement of gene expression. Individual
points indicate biological replicates (average of 3 technical replicates). All numeric data points
in S2 Data.
(PDF)

**S3 Fig. Growth curves for regulatory RNA mutants.** Growth curves in a complete synthetic
medium (CDM) in the presence and absence of ligand. All curves represent at least two biolog-
ical replicates. Error bars represent standard error across all replicates (n = 5–9). Doubling
time and carrying capacity measurements extracted from each individual curve under each
condition are indicated on graphs below. In each condition mutants were compared to the
WT$^C$ via Kruskal-Wallis test followed by Dunn's test of multiple comparisons, those displaying
significant changes are indicated ($^*p<0.05$, $^{**}p<0.01$, $^{***}p<0.001$, $^{****}p<0.0001$) (**A**)
0716_TPP mutants grown in the presence and absence of thiamine. (**B**) 0719_TPP mutants
grown in the presence and absence of thiamine. (**C**) 0726_TPP mutants grown in the presence
and absence of thiamine. (**D**) 2199_TPP mutants grown in the presence and absence of thia-
mine. (**E**) 0178_FMN mutants grown in the presence and absence of riboflavin#. (**F**)
0488_FMN mutants grown in the presence and absence of riboflavin#. (**G**) 0701_pyrR regula-
tor mutants grown in the presence and absence of uracil#. (**H**) 1286_pyrR mutants grown in
the presence and absence of uracil#. (**I**) 0408_Glycine mutants grown in the presence and

absence glycine. **(J)** 1069_TrpT-box mutants grown in the presence and absence tryptophan. **(K)** 1847_Guanine mutants grown in the presence and absence of guanine. **(L)** Parameters extracted from previously published growth curves for the 1278_pyrR mutants [64] **(M)** Growth curves in a semi-defined minimal media (SDMM) for 0178_FMN_WT$^C$ and mutants showing autolysis phase of characteristic of *S. pneumoniae* growth in richer medium. #These growth curves are also shown in Figs 6 or 7, but repeated here for accessibility to the entire data set. All numeric data points in S3 Data.
(PDF)

**S4 Fig. Summarized relative fitness and relative growth parameters for individual mutant strains.** The fitness mean of each individual strain is displayed for nasopharynx colonization **(A)**, lung infection **(B)**, and transition to blood models **(C)**. Orange circles = mLI (Ligand Insensitive), green triangle = mON (constitutively active gene expression), and red square = mOFF (repressed gene expression), error bars correspond to the standard deviation (S4 Table). Filled points correspond to strains with a statistically significant change (p. adj. <0.05) from the *in vitro* competition (rich medium) control (Kruskal-Wallis test followed by Dunn's test of multiple comparisons, adj. p<0.05). Open points are not significantly distinct from the *in vitro* control competition. The black-filled point (1286_pyrR_mON) corresponds to an environment under which none of the mutant strain was recovered despite repeated attempts, indicating a fitness close to 0. Graphs representing individual mouse competitions for each strain are found in S5 Fig. Relative carrying capacity **(D,E)** and relative growth rate **(F,G)** for mutant strains compared to the WT$^C$ strain. Points represent the mean of 5–9 replicates, and the error bars represent the standard deviation (S3 Table). Some error bars are smaller than the size of the point and therefore not visible. Open points are not statistically significantly different from the WT$^C$ strain. Colored points represent values that are statistically significant from the WT$^C$ strain (Kruskal-Wallis test followed by Dunn's multiple comparisons test, adj. p <0.05, S4 Fig and S3 Table). Vertical lines separate groups of cis-acting regulators interacting with the same ligand. All numeric data points in S2 Data.
(PDF)

**S5 Fig. Fitness values for individual mouse lung infections and nasopharynx colonization assays compared to the *in vitro* control in semi-defined or rich medium.** Each data point consists of a fitness value determined from a single mouse infection, or *in vitro* competition in rich medium. Statistical difference from the control experiment determined using the Kruskal-Wallis test followed by Dunn's test of multiple comparisons. (* = adj. p<0.05, ** adj. p<0.01, *** p < .001, **** adj. p < .0001). **(A)** 0719_TPP WT$^C$, mLI, mON and mOFF. **(B)** #0178_FMN WT$^C$, mLI, mON and mOFF. **(C)** #0488_FMN WT$^C$, mLI, mON and mOFF. **(D)** #0701_pyrR WT$^C$, mLI, mON and mOFF. **(E)** #1286_pyrR WT$^C$, mLI, and mON (1286_pyrR_mOFF mutant not successfully constructed). **(F)** 0408_Glycine WT$^C$, mLI, mON and mOFF. **(G)** 1069_Trp-Tbox WT$^C$, mLI, mON and mOFF. **(H)** 1847_Guanine WT$^C$, mLI, mON and mOFF. #These graphs are also shown in Figs 6 or 7, but repeated here for accessibility to the entire data set. All numeric data points in S2 Data.
(PDF)

**S6 Fig. Modelling co-culture indicates infection fitness defects cannot be directly attributed to planktonic growth parameters. (A)** Fitness modelled using a variety of parameter sets as indicated to demonstrate sensitivity of the modelled fitness to parameters. **(B)** Modelled fitness during exponential phase (120 minutes) and **(C)** stationary phase (500 minutes) for all RNA regulator mutants based on their relative growth rate and carrying capacity in CDM lacking the target nutrient as reported on S3 Table. Modelled fitness calculated based on a co-

culture model with the growth rate of the reference strain $r_a$ = 0.0252 and carrying capacity $K_a$ = 0.462. Growth rate and carrying capacity of test strains are scaled by the relative growth rate and carrying capacity such that $r_b = r_a*$relative growth rate, and $K_b = K_a*$relative carrying capacity on S3 Table. Horizontal line drawn at fitness $W$ = 1 for visual reference. Error was estimated by repeating calculations with values of $r_a$ and $K_a$ adjusted by the standard deviations of the mean reported on S3 Table. **(D)** Negative control *in vitro* competitions conducted between *S. pneumoniae* TIGR4 and 0701_pyrR_WT[C] or 1278_pyrR_WT[C] show no significant change in fitness. Bars represent mean fitness and error bars standard deviation for individual biological replicates shown as points. All numeric data points in S2 Data.
(PDF)

**S7 Fig. Modelled fitness correlates weakly with measured fitness from *in vivo* environments compared to correlations between *in vivo* fitness measures. (A-C)** Modelled stationary phase fitness vs. mean measured fitness in the nasopharynx, lung, and blood. **(D-F)** Modelled exponential phase fitness vs. mean measured fitness in the nasopharynx, lung, and blood. **(G-H)** Comparison of the three different *in vivo* environments shows significant correlation in fitness values. Pearson's correlation coefficient and Bonferroni adjusted p-value (N = 9) are reported for each comparison. Error bars correspond to error bars reported in S4A–S4C, and S6B, S6C Figs. All numeric data points in S2 Data.
(PDF)

**S1 Data. All numerical data points for main text figures.**
(XLSX)

**S2 Data. All numerical data points for S2, S4, S5, S6, and S7 Figs.**
(XLSX)

**S3 Data. All numerical data points and raw data for S3 Fig.**
(XLSX)

## Acknowledgments

We would like to thank Brooke Hensley, Hadley Johnson, and Maya Cooley for their help with confirming regulator activities using Miller Assays, Maribel Andrade and Rebecca Korn for the SP_0178_KO strain, Quinlan Furumo for help with data sampling and statistical analysis in R, as well as Arianne Babina and Babak Momeni for helpful discussions regarding data analysis and interpretation, and Daniel Beringer for careful copyediting of the manuscript.

## Author Contributions

**Conceptualization:** Indu Warrier, Tim van Opijnen, Michelle M. Meyer.

**Data curation:** Indu Warrier, Ariana Perry, Sara M. Hubbell, Matthew Eichelman, Michelle M. Meyer.

**Funding acquisition:** Tim van Opijnen, Michelle M. Meyer.

**Investigation:** Indu Warrier, Ariana Perry, Sara M. Hubbell.

**Project administration:** Michelle M. Meyer.

**Supervision:** Tim van Opijnen, Michelle M. Meyer.

**Visualization:** Indu Warrier.

**Writing – original draft:** Michelle M. Meyer.

**Writing – review & editing:** Indu Warrier, Ariana Perry, Sara M. Hubbell, Matthew Eichelman, Tim van Opijnen.

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
