## [Decision Letter · Decision Letter 0]

8 Jan 2024

Dear Dr Meyer,

Thank you very much for submitting your Research Article entitled 'RNA cis-regulators are important for *Streptococcus pneumoniae in vivo * success' to PLOS Genetics.

The manuscript was fully evaluated at the editorial level and by independent peer reviewers. The reviewers appreciated the attention to an important topic but identified some concerns that we ask you address in a revised manuscript.

We therefore ask you to modify the manuscript according to the review recommendations. Your revisions should address the specific points made by each reviewer.

Yours sincerely,

Kai Papenfort

Academic Editor

PLOS Genetics

Sean Crosson

Section Editor

PLOS Genetics

Dear Dr. Meyer.

Thank you again for submitting your work to PLOS Genetics. We have now received comments from the reviewers of your manuscript. Overall, all three reviewers were very positive about your manuscript, however, they also requested several changes to the text, as well as few additional experiments.

Please make sure to respond to all their comments in your revised manuscript and the rebuttal letter.

Best wishes,

Kai Papenfort

(Academic Editor)

Reviewer's Responses to Questions

**Comments to the Authors:**

Reviewer #1: Summary:

In the manuscript entitled "RNA cis-regulators are important for Streptococcus pneumoniae in vivo success" submitted to PLOS genetics, Warrier and colleagues investigated how altering the functionality of 11 different riboswitches that respond to particular metabolites impact S. pneumoniae growth in chemically defined medium (CDM) and fitness in the blood, lungs, and nasopharynx of mice. Briefly, the authors mutate riboswitches that sense riboflavin, uracil, glycine, TPP, guanine, and charged tryptophan tRNAs levels so that they are no longer or less responsive to the metabolite, locked in an on-state, or always off. The authors first validate that the mutations have their intended effect, which do so in most cases (Figs. 1 and S2). The authors then determine the growth rate and yield for these strains in CDM (Figs. 2, S3, and S4), and fitness in the lungs, blood, and nasopharynx of mice (Figs. 3, S4, and S5) compared to a strain with the wild type riboswitch preserved. Overall, the authors find collectively that locking these riboswitches in the off position through mutation leads to S. pneumoniae growth defects in CDM particularly in the absence of its cognate metabolite, but the locked-off mutations as well as the less responsive and locked on mutations tend to cause a more substantial defect in S. pneumoniae fitness in vivo than in vitro (Figs. 2, 3, and 4). Overall, the authors find that disrupting riboswitches that sense uracil or riboflavin levels tend to have a more substantial effect on in vitro growth and in vivo fitness (Figs. 6 and 7), but find relatively poor correlation between the effects of mutations on growth rate or yield and in vivo fitness (Fig. 5).

In my opinion, the authors have done an impressive amount of work to establish the contribution of metabolite sensing riboswitches to the growth and fitness of S. pneumoniae. Moreover, they have demonstrated that gene regulation by riboswitches has a much more profound impact on the ability of S. pneumoniae to survive and thrive in the in vivo environments tested than in the vitro conditions examined. I have no substantial criticisms to offer regarding this work. I do have some suggested corrections to typographical errors that I found.

Major Criticisms:

None.

Minor Concerns:

L19. Change to "a rapid response"

L40. Change to "development of"

L189. Change to "We found that"

Figure 2D and 2E. For clarity, the authors should add the term "Relative" to "Carrying Capacity" or "Growth Rate" at the top of the graph.

L411. Change to "observed"

L463. (Fig. S3L) is incorrectly referenced here.

L508. FigS3L should be referenced along with Fig. S7B.

K528 Change to "Fig. 7G". also reference Fig. S3L.

Reviewer #2: The review was uploaded as an attachment.

Reviewer #3: In their study, Warrier et al analyzed the impact of 12 highly conserved RNA cis-regulators inhibited in presence of various metabolites in in vitro and in vivo growth of Streptococcus pneumoniae. They introduced specific mutations either to perturb ligands binding (mLI), or to constitutively activate (mON) and repress (mOFF) expression. The key result was that miss-regulation (positive or negative) induced a stronger fitness decrease in mouse infection than in growth medium for 75% of the mutant strains. The authors emphasized their findings with two specific examples: FMN and uracil responsive RNA elements. In brief, the work was original, the experiments were well conducted, and the results were clearly presented. Below are few comments and suggestions:

Because all the riboswitches were repressed in presence of ligands (Fig 1A) and mLI/mON mutations presented a greater activity than the WT in vitro (Fig 1D and comment below), it is difficult to assume whether the RNA regulators were ON or OFF in vivo. I would suggest that the authors check the relative expression of one of the WT RNA regulators, and its derivatives in organs by qRT-PCR. Furthermore, the authors should discuss the apparent paradox that the mON and mOFF mutants behaved similarly in infection models.

L674: “50S ribosomal protein gene”, change to “L33 50S ribosomal protein gene”

L221-224: what are the authors hypotheses about the increase expression of the mON and mLI mutants? This should be discussed.

From L506: the paragraph is very confusing. The authors referred to in vivo results for the 1278_pyrR-mOFF mutant that I could not find. The legend of Figure S7 is not clear: to which RNA regulator does it refer to?

**Have all data underlying the figures and results presented in the manuscript been provided?**

Reviewer #1: Yes

Reviewer #2: Yes

Reviewer #3: Yes

PLOS authors have the option to publish the peer review history of their article (what does this mean?). If published, this will include your full peer review and any attached files.

Reviewer #1: No

Reviewer #2: No

Reviewer #3: No

---

## [Decision Letter · Decision Letter 1]

19 Feb 2024

Dear Dr Meyer,

We are pleased to inform you that your manuscript entitled "RNA cis-regulators are important for *Streptococcus pneumoniae in vivo * success" has been editorially accepted for publication in PLOS Genetics. Congratulations!

Yours sincerely,

Kai Papenfort

Academic Editor

PLOS Genetics

Sean Crosson

Section Editor

PLOS Genetics

Comments from the reviewers (if applicable):

Dear Dr Meyer, dear Michelle.

I have now received feedback from the referees on your revised manuscript and I am happy to inform you that all three suggested to accept it (and so have I).

Please note that referees #1 and #2 have some additional minor comments (see below) that need to be fixed before publication.

Congratulations and all the best

Kai Papenfort

Reviewer's Responses to Questions

**Comments to the Authors:**

Reviewer #1: Summary

In the revised manuscript submitted to PLOS Genetics entitled "RNA cis-regulators are important for Streptococcus pneumoniae in vivo success", Warrier and colleagues evaluated the fitness consequences for S. pneumoniae in vitro (in liquid culture) and in vivo (in a mouse) of mutating each of 12 riboswitches so that they are ligand insensitive or locked in a constitutively active or repressed state. In short, the authors find that altering these riboswitches to be ligand insensitive or constitutively active in most cases has no impact on S. pneumoniae carrying capacity or growth rate, whereas strains harboring the constitutively repressed version of the riboswitch frequently had a defect in these growth parameters particularly in the absence of the ligand (Fig. 2). In contrast, mutations that cause the riboswitch to be ligand insensitive, constitutively active, or constitutively repressed tended to cause defects in S. pneumoniae in vivo fitness in the nasopharynx, lungs, and/or blood (Fig. 3). Furthermore, the authors provided evidence that whether or not a mutation in a riboswitch caused growth of S. pneumoniae to be defective in vitro was not very predictive of whether that mutation caused a deficiency in S. pneumoniae fitness in vivo (Figs. 4 and 5). Finally, the authors compare how mutating different FMN (Fig 6) and pyrimidine riboswitches impacted S. pneumoniae growth in vitro and fitness in vivo.

As I stated previously, the authors have done an impressive amount of work to establish the contribution of metabolite sensing riboswitches to the growth and fitness of S. pneumoniae. Moreover, they have demonstrated that gene regulation by riboswitches has a much more profound impact on the ability of S. pneumoniae to survive and thrive in vivo than in vitro. I still do not have any substantial criticisms to offer. Some trivial typographical errors indicated below should be corrected.

Major Concerns:

None

Minor comments:

P2 L24. Change to "constitutively active or repressed"

P2 L27. Change to "strains with either constitutively active or repressed gene expression"

P5 L91. Change to "seem to be to accelerate a return"

P6 L106. Change to "displaying either constitutively active or repressed gene"

P7 L126. Change to "activated or repressed"

P9 L188. Change to "qRT-PCR"

P10 L204. Change to "or truncate the aptamer"

P10 L210. Change to "qRT-PCR"

P11 L215. After "constructs" cite Fig. S2.

P13 L258. Delete the word "ranging"

P25 L528. Change "7B" to "7C"

P26 L549. Change "7G" to "7H"

P30 L635. Change to "shown to be essential"

P30 L645. Change to "intact pathway for tryptophan"

P49, L1108. Change to qRT-PCR

P49, L1116. Change to "suggest that they"

P50, L1127. "activities, and we retained this mutant."

P53, L 1202. insert a space between "." and "Figure"

Reviewer #2: The authors were very responsive to the reviewer's comments. The manuscript is improved and it appears ready for publication. Only a few typos remain:

Line 255. to [the] regulatory sequence.

Line 371. = .0035 should be 0.0035

Line 755 12.5 hours [were] used for

Reviewer #3: I think the authors have convincingly replied to all my comments.

**Have all data underlying the figures and results presented in the manuscript been provided?**

Reviewer #1: Yes

Reviewer #2: Yes

Reviewer #3: Yes

PLOS authors have the option to publish the peer review history of their article (what does this mean?). If published, this will include your full peer review and any attached files.

Reviewer #1: No

Reviewer #2: No

Reviewer #3: No

**Data Deposition**

http://datadryad.org/submit?journalID=pgenetics&manu=PGENETICS-D-23-01386R1

**Press Queries**

---

## [Editor Report · Acceptance letter]

29 Feb 2024

PGENETICS-D-23-01386R1 

RNA cis-regulators are important for *Streptococcus pneumoniae in vivo * success 

Dear Dr Meyer, 

We are pleased to inform you that your manuscript entitled "RNA cis-regulators are important for *Streptococcus pneumoniae in vivo * success" has been formally accepted for publication in PLOS Genetics! Your manuscript is now with our production department and you will be notified of the publication date in due course.

With kind regards,

Anita Estes

PLOS Genetics

On behalf of:
